# Calculation of the SPI, SPEI, and GRDI Indices for Historical Climatic Data from Doñana National Park: Forecasting Climatic Series (2030–2059) Using Two Climatic Scenarios RCP 4.5 and RCP 8.5 by IPCC

María José Montes-Vega [1], Carolina Guardiola-Albert [2] and Miguel Rodríguez-Rodríguez [1,*]

[1] Department of Physical, Chemical and Natural Systems, Pablo de Olavide University, Utrera Road Km 1, 41013 Seville, Spain; mjmonveg@upo.es

[2] Spanish Geological Survey-CSIC, Rios Rosas 23, 28003 Madrid, Spain; c.guardiola@igme.es

\* Correspondence: mrodrod@upo.es

**Abstract:** In this study, we utilized three different indices to assess drought conditions in the Doñana National Park (DNP) located in southern Spain. These indices included the Standardized Precipitation Index (SPI), which is based on precipitation statistics, the Standardized Precipitation Evapotranspiration Index (SPEI), which incorporates both precipitation and air temperature data, and the Groundwater Recharge Drought Index (GRDI), a newly developed index specifically designed to evaluate groundwater drought. The analysis covered the time period from 1985 to 2015, and future projections were made for the years 2030–2060 under different climate scenarios (RCP 4.5 and RCP 8.5). Our findings revealed a significant decrease in total precipitation of approximately 13–14% compared to historical records (1985–2015). Moreover, severely to extremely wet periods exhibited a reduction ranging from 25% to 38%. A key contribution of this study is the application of the GRDI index, which allowed us to assess groundwater recharge rates. We observed a decline in the simulated mean recharge rates during the 21st century when compared to the historical period spanning from 1950 to 2009. This decline can be attributed to increased evapotranspiration. The results of this research provide valuable insights for the Spanish water resources administration. The observed reductions in precipitation and groundwater recharge rates emphasize the need for appropriate mitigation measures. The findings will aid the administration in formulating an integrated water resources management strategy in the Doñana National Park and its surrounding basin. By understanding the projected changes in drought conditions, the administration can make informed decisions to ensure sustainable water resource management in the region.

**Keywords:** groundwater; climate change; coastal aquifers; drought; recharge; SPI; SPEI; GRDI; future projections; environmental risks

## 1. Introduction

The sustainable management of water resources in the context of global change is a complex task that needs to be accomplished using multiple approaches. The urban, industrial, and agricultural use of water and their relationship with food, energy, and biodiversity must be quantified with precise hydrological tools as, for example, groundwater modeling and/or the development of new hydrological indexes, to allow a better understanding of the relative importance of specific hydrological processes. Densely populated or agricultural coastal areas, where access to fresh surface water commonly is limited, are largely dependent on groundwater. In a semi-arid climate, where surface water is scarce, and its availability is intermittent, the use of groundwater resources constitutes more than 80% of the total water use for urban and irrigation demand [1]. On the other hand, groundwater is essential for the maintenance of dependent ecosystems, such as outstanding universal

value (OUV) ponds and lagoons. To be deemed of outstanding universal value, a property must also meet the conditions of integrity and/or authenticity and must have an adequate protection and management system to ensure its safeguarding, as stated in the Operational Guidelines for the Implementation of the World Heritage Convention.

It is well known that the demand for groundwater as a safe alternative to surface water will increase because of rapid population development, growing demand for food and urbanization, and climate change. Climate change is expected to alter drought occurrence and intensity. Increased drought duration in many parts of the world will further challenge drought mitigation strategies [1]. In fact, drought is one of the major causes of agricultural, economic, and environmental damage [2]. It remains difficult to determine the onset, intensity, and duration of drought in spite of a great number of studies committed to this effort [2–4]. One of the significant challenges in drought studies is the limited availability of long-term climatic datasets, which are crucial for the development of robust climatic models, including drought predictive models. In recent decades, numerous efforts have been made to develop and enhance drought indices. Among these, the standardized precipitation index (SPI) has emerged as a valuable tool for assessing drought dynamics in a specific region [4]. Despite being relatively effective, this indicator has several flaws [2]. One of the problems of using the SPI as a drought indicator is that its calculation is exclusively based on precipitation data. Other variables associated with drought, such as air temperature, are not included. The standardized precipitation evapotranspiration index (SPEI) is based on precipitation and air temperature data [2]. The calculation of this index uses the accumulated differences between precipitation and potential evapotranspiration (see Section 2 below). Therefore, it is an excellent method to model the effects of global warming on drought situations.

Several studies were conducted for the sustainable management of groundwater in the context of global change, using meteorological, agricultural, and hydrological indexes. The European Groundwater Drought Initiative, a pan-European collaboration, has recently highlighted the major gaps in European drought research capability related to groundwater droughts. Among the most important research performed in this area, an index was developed [1] to determine groundwater drought, i.e., the continuous and extensive occurrence of the below average availability of groundwater. The index was named Groundwater Recharge Drought Index (GRDI). One of the main results of this work was that they were able to separate the human effects on the hydrological system from the effects of natural drought. The application of this index could also be applied to assess the sustainable management of coastal aquifers.

In a recent risk assessment of the ecological damage due to groundwater abstraction in the study area of the present work, Doñana National Park (henceforth, DNP), one of the main conclusions was that appropriate indicators to study the hydrological context of the detrital aquifer are urgently needed [5]. Moreover, the information that we complied from EUROCODEX project for the Doñana area is quite in agreement with the studies performed over all the Guadalquivir catchment for climate change scenarios (i.e., RCP 4.5 or RCP 8.5 scenario from IPCC panel).

The authors considered that the level of scientific uncertainty is unacceptable for an iconic World Heritage site such as the DNP. To avoid any potential risk to the groundwater-dependent OUV ecosystems within DNP, it is highly recommended that the evolution of the groundwater recharge in the aquifer is studied in the future, and that an integrated water resources management is implemented. A common practice to assess this issue is to analyze future climate projections downscaled from global climate change models.

With the given suggestion in mind, the primary goals of this paper are as follows:

1. Incorporate historical climatic data from DNP to enhance the SPI, SPEI, and GRDI models.
2. Contrast these indices with those employed by the managers of DNP.
3. Utilize these indices to predict climatic patterns (2030–2059) under two climatic scenarios, namely RCP 4.5 and RCP 8.5 by IPCC.

### 1.1. Study Site

#### 1.1.1. Climate

The DNP experiences a Mediterranean climate with hot, dry summers and mild, wet winters. During the summer months, temperatures can soar to over 40 °C while winter temperatures typically range from 5 to 15 °C. The study area receives most of its rainfall in the winter months, with an average (1970–2022 period) of 525 mm of precipitation per year. These weather conditions create a diverse landscape in Doñana, with a mix of wetlands, dunes, and forests.

#### 1.1.2. Hydrogeology

The aeolian littoral mantle of Doñana, where we focused this study, is an unconfined dune aquifer that forms part of the Almonte-Marismas detrital aquifer system (Figure 1), in southwest Spain. This area is located between the marshes and the sea, where many coastal ponds, situated in the dune slacks, are supported. These dune-slack ponds are mostly temporary and are directly supplied by rainfall and groundwater [5,6]. In the case of Santa Olalla pond, which is a permanent coastal pond, only 20% of the water inputs comes from direct rainfall and 80% from groundwater discharge [7].

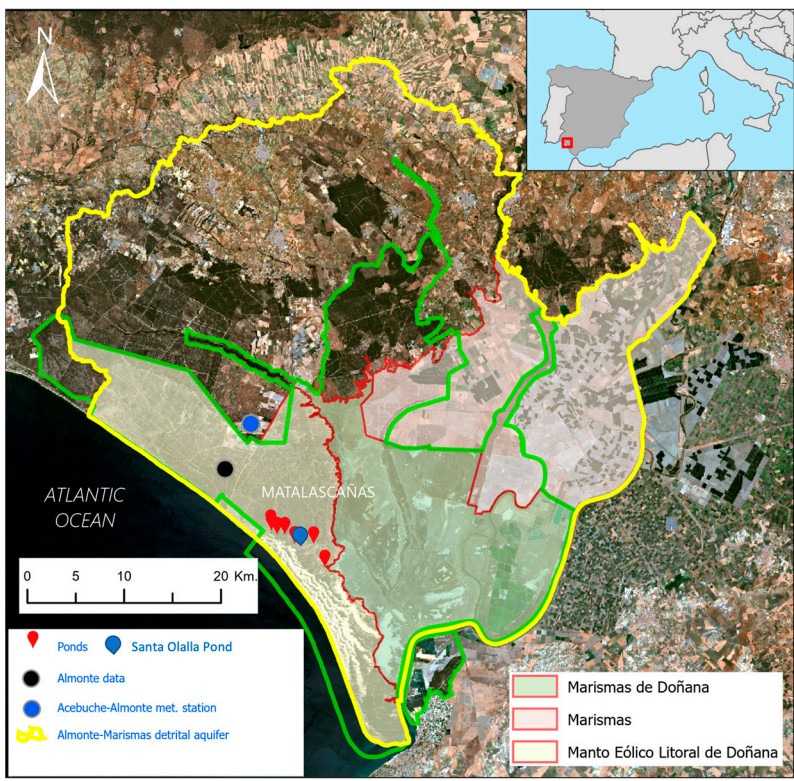

**Figure 1.** Location of the study site. The blue dot shows the meteorological station (Almonte-Acebuche). The black dot shows the location (longitude −6.54, latitude 37.09) of the zone where the future projections are based on, within the Manto Eólico Litoral de Doñana groundwater body (shaded in yellow). Green line represents the border of Doñana National Park.

The hydroperiod of the dune ponds is determined by the discharge coming from the aeolian sand aquifer, which is recharged by rainfall. Some of the ponds were significantly affected by the lowering of the groundwater levels in the aeolian aquifer due to changes in rainfall and to pumping for public water supply to Matalascañas [7]. There is a hydrological connection between the aeolian sand aquifer and deeper regional detrital aquifer, which also influences the pond's hydrology. The aeolian system developed continuously from the beginning of the formation, first covering the alluvial sediments of the Almonte Formation and then non-lapping over the lateral equivalents of the Lebrija and Marismas

formations [5]. Depending on the location and altitude of the ponds, their hydrology is determined to a greater or lesser extent by both the local groundwater fluxes within the aeolian sand aquifer and the regional groundwater levels [8].

### 1.1.3. Water Management

Doñana is an area enclosing a variety of conflicting water resource demands required to maintain agriculture, industry, and tourism [9]. The Confederación Hidrográfica del Guadalquivir (CHG) is the administrative organization responsible for managing water resources in the Guadalquivir river basin in southern Spain, where the study area is encompassed. In 2020, CHG officially declared that 3 of the 6 groundwater bodies in Doñana were overexploited. In recent years, the region has been affected by prolonged droughts, which have had a significant impact on agriculture, tourism, and the environment. To address this issue, the CHG developed a drought management plan that aims to improve the resilience of the region's water resources. This plan includes measures such as the construction of the new hydraulic infrastructure, the promotion of water conservation practices, and the implementation of more efficient irrigation systems. Additionally, the CHG is working with local stakeholders, such as farmers and tourism operators, to raise awareness about the importance of water conservation and to develop strategies for adapting to the changing climate. While the plan is still in its early stages, it is expected to represent a crucial step towards ensuring the long-term sustainability of water resources in the Guadalquivir river basin.

## 2. Materials and Methods

To accomplish this study, two drought indices (SPI and SPEI) and one recharge index (GRDI) were calculated for observed and future climate data. In the following sections, such indices are described.

### 2.1. SPI

The Standardized Precipitation Index (SPI) is a drought index primarily based on precipitation [4]. It quantifies precipitation deficiency by comparing observed precipitation to the long-term average, and then normalizing it using the standard deviation of the historical precipitation record. The SPI serves as a standardized measure to assess precipitation anomalies, enabling the comparison of drought severity across different regions and timeframes. The SPI can be computed for various time scales, ranging from very short-term (such as daily or weekly) to considerably longer-term (such as 12 months or more). The choice of time scale depends on the specific application and temporal characteristics of the phenomenon under investigation. Generally, shorter time scales are more suitable for monitoring and predicting short-term droughts, while longer time scales are better for assessing long-term climate variability and trends.

In this study, we chose the SPI-12, which represents the precipitation anomalies over the preceding 12 months. This time scale is often employed to evaluate long-term drought conditions since it captures the cumulative impact of precipitation deficits over an extended period. In any case, as a previous step of this investigation, we calculated SPI-6 (6 months) and analyzed the results obtained. This time-step is better suited as an indicator for reduced stream flow and reservoir storage. Moreover, this is the time scale chosen by CHG to manage droughts in the drought management plan of the Doñana area. Therefore, three of the great strengths of the SPI drought index are its simplicity, flexibility, and temporal versatility. However, this index has limitations because precipitation is the only meteorological factor studied to describe a multifactorial phenomenon such as drought. For the short series of precipitation, the uncertainty increases, and the SPI tends to underestimate the intensity of the drought or wet period, so it tends to give smoother drought results. This reason, and the fact that temperatures are not considered in its estimate, make the use of the SPI in climate studies questionable [10,11].

As shown in Table 1, the results of the SPI were classified, into seven different categories based on the initial classification of [4]. Droughts periods occur when SPI has a continued negative value, from −1 or less. For the present study, we considered that a wet period starts when the value of the index become positive [12].

**Table 1.** Classification of drought severity for the Standardized Precipitation Index (SPI) following [12].

| Category | SPI | Color Assigned |
|---|---|---|
| Extremely wet | 2 and above | |
| Severely wet | From 1.5 to 1.99 | |
| Moderately wet | From 1 to 1.49 | |
| Near normal | From −0.99 to 0.99 | |
| Moderate drought | From −1 to −1.49 | |
| Severe drought | From −1.5 to −1.99 | |
| Extreme drought | −2 and less | |

### 2.2. SPEI

The Standardized Precipitation Evapotranspiration Index (SPEI) is another drought index based on the precipitation, maximum and minimum temperature data. This index is calculated similarly to the SPI, but using the accumulated difference between precipitation and potential evapotranspiration. With this easy water balance, the SPEI can be also calculated at 6 months (SPEI-6) and 12 months (SPEI-12) [2].

Using both indices, it can be observed and compared whether periods of drought are influenced by increases in temperature in periods when there are no major changes in rainfall [13].

The SPEI, due to its characteristics, is an index especially used in studies related to the effect of global warming on drought and drought monitoring. In this study, the "SPEI" package in R-Studio 4.0.3 software was used to calculate both drought indices, SPI and SPEI for a 12-month scale [14,15].

### 2.3. GRDI

The Groundwater Recharge Drought Index (GRDI), which is determined based on the values of aquifer precipitation recharge, was calculated following the protocol of Goodarzi [1]. This method is inspired by the SPI approach and follows the same steps used for precipitation analysis, but applied to the recharge time series instead of precipitation data. Because the GRDI calculation is based on groundwater recharge, the uncertainty in recharge estimation may reduce the accuracy of this index [1]. For the present GRDI computations, the recharge time series was calculated using a soil water balance computing program [16]. The code, called TRASERO V2.3, was applied to compute the recharge at a daily time step and then upscaled to obtain month recharge values. The input data needed for the recharge assessment are rainfall and maximum and minimum temperature at a daily time step. Classic Thornthwaite method was used for effective rainfall calculation. The Thornthwaite method is a widely used approach for estimating effective rainfall, which refers to the amount of precipitation available for soil moisture recharge and plant growth. At a daily scale, the effective rainfall calculation through the Thornthwaite method involves several steps. Firstly, the potential evapotranspiration (PET) is estimated using the Hargreaves equation, which takes into account the maximum and minimum daily temperature data and the latitude of the location. The PET represents the amount of water that would evaporate and transpire under optimal moisture conditions. Next, the field capacity (i.e., maximum soil moisture for the site) is needed as a site-specific parameter. The field capacity, in mm, helps determine the water deficit or surplus for a given day. By summing the daily rainfall values, an estimate of the actual evapotranspiration (AET) is obtained. The effective rainfall, also in mm, can then be determined by subtracting the AET from the total rainfall. This approach provides valuable insights into the water availability and helps manage irrigation and water resources at a daily scale. To consider the

uncertainties related with field capacity, two values were used in the recharge computation, namely 40 and 70 mm. The runoff threshold was assigned based on tabulated values that depend on the soil type, vegetation cover, hydrological characteristics, and land uses. We used the same ranges as those used in Naranjo-Fernández et al. (2020) for the aeolian sand zone, in which the runoff threshold for each sub-area was estimated and then the direct runoff values were determined [6]. Infiltration assessment was performed by the Soil Conservation S (SCS) method. A runoff threshold (Po), which depends on the soils' characteristics and the land use, must be introduced. To assign it, the program provides a query in which the user can choose one of these two parameters according to the characteristics of the soils and land use. As a result, a runoff threshold (Po) of 90 mm was used.

Once the recharge values were obtained for the study area, we verified that the data had a good fit to the gamma distribution from the *p*-value using the Kolmogorov–Smirnov statistical test [1]. The same "SPEI" package in R-Studio 4.0.3 software was used to calculate GRDI values for a 12-month scale. To use this package, instead of inserting precipitation as the input time series, we used the recharge values obtained by the implementation of the TRASERO code at a daily scale and then upscaled them to obtain month recharge values.

*2.4. Data*

Rainfall and maximum/minimum temperature daily data were obtained from the Almonte-Acebuche meteorological station, which is operated by the Spanish State Meteorological Agency. The historical time series of observations spanned from 1 October 1975 to 30 September 2017, following the hydrological year. To facilitate the analysis, the daily data were aggregated into monthly values. The R programming language, specifically the SPEI package, was utilized for further analysis and calculations [15], whilst potential evapotranspiration (PET) data were calculated using the Hargreaves equation. The Hargreaves equation is a temperature-based method which is used as a representative expression for the daily values of PET:

$$\text{PET (mm/day)} = 0.0135 \, (\text{tave} + 17.78) \, \text{Rs}$$

where:

$$\text{tave} = \text{average temperature, } ^\circ\text{C}$$

$$\text{Rs} = \text{incident solar radiation, mm/day}$$

$$\text{Rs} = \text{R0KT} \, (t_{max} - t_{min})^{0.5}$$

$$\text{R0} = \text{extraterrestrial solar radiation, MJules/m}^2/\text{day}$$

$$\text{KT} = \text{empirical coefficient (0.162 for interior and 0.19 for coastal areas)}$$

$$t_{max} = \text{maximum temperature, } ^\circ\text{C}$$

$$t_{min} = \text{minimum temperature, } ^\circ\text{C}$$

Daily precipitation as well as the maximum and minimum temperature data from the years 2030–2059 were acquired for the upcoming climate series. These data were obtained from the Spanish Climate Change Office's platform on adaptation to climate change (http://adaptecca.es/en (accessed on 23 January 2023)), which sources its data from the Spanish State Agency of Meteorology [17]. These data have undergone small-scale dynamic downscaled processing. This approach is preferred for our study as it provides a physically consistent scaling between variables, such as rainfall and temperature, unlike statistical methods. The data were sourced from the EUROCORDEX project [18], which is a comprehensive initiative aimed at generating high-resolution climate projections specifically tailored to the European region. The project is highly regarded for its scientific reliability due to its rigorous approach to modeling and analysis, as well as its adherence to established scientific standards and protocols. The models used in EURO-CORDEX have

undergone extensive validation and testing, and the project was peer-reviewed by leading experts in the field. In addition, the project team follows strict quality control procedures to ensure the accuracy and consistency of its data. Overall, the EURO-CORDEX project is widely recognized as a valuable resource for water policymakers, researchers, and other stakeholders seeking to better understand the potential impacts of climate change on the European continent. We used 16 different models for RCP 4.5 and RCP 8.5 scenarios from the IPCC for the closest area to our study site (see Figure 1). However, these data cannot be used directly, due to the biases they present when compared with the observed data. For this, they need to be calibrated using a bias correction technique [19]. We applied this correction to 16 different models for each scenario, and then we ensembled the results to calculate SPI, SPEI, and GRDI for future simulations in both scenarios (RCP 4.5 and 8.5). The ensemble was made with the mean of the 16 models.

## 3. Results

### 3.1. Historical Data

This section analyzes the indices for historical data observed at the meteorological station (Table 2). We calculated the SPI, SPEI, and GRDI indices for 12 months, so the first results that we obtained in each index are for September 1976 (data series beginning in October 1975).

**Table 2.** Statistical resume for the observed historical time series (1975–2017) based on hydrological year for precipitation and maximum–minimum temperature variables.

| Data | Max | Min | Mean |
|---|---|---|---|
| Precipitation (mm) | 1003.95 | 208.57 | 564.12 |
| Maximum temperature (°C) | 25.64 | 22.65 | 23.97 |
| Minimum temperature (°C) | 12.43 | 4.62 | 9.78 |

Figure 2 illustrates the temporal progression of the Standardized Precipitation Index (SPI) and the Standardized Precipitation–Evapotranspiration Index (SPEI). Notably, three prolonged drought periods consisting of four to five consecutive dry years can be observed: 1980–1984, 1991–1995, and 2014–2017. Additionally, there are two instances of two consecutive dry years: 2005–2006, along with two remarkably dry years in 1999 and 2012. The SPI and SPEI indices exhibit similar patterns, although the SPEI tends to display less extreme values compared to the SPI. It appears that dry periods are relatively mitigated in comparison to wet periods. This phenomenon can be attributed to the consideration of evapotranspiration in the SPEI calculation, which has a greater impact on dry periods. Shifting focus to wet periods, there are two four-year periods (1987–1990 and 2001–2004), one three-year period (1996–1998), and two biennial periods (1976–1977 and 2010–2011). On the graphs of the classifications following Table 1 (Figure 2B,D), we can discern different types of wet and dry periods along the time. Extreme drought months only accumulate during four months in 2005, once in the 42-year study period, and only for SPI. Severe droughts are occur more frequently in consequent months, up to seven months in 1999 for SPI. Comparing SPI with SPEI indices, one month of extreme drought (November 1981) and one extremely wet month (December 1996) coincide on both indices.

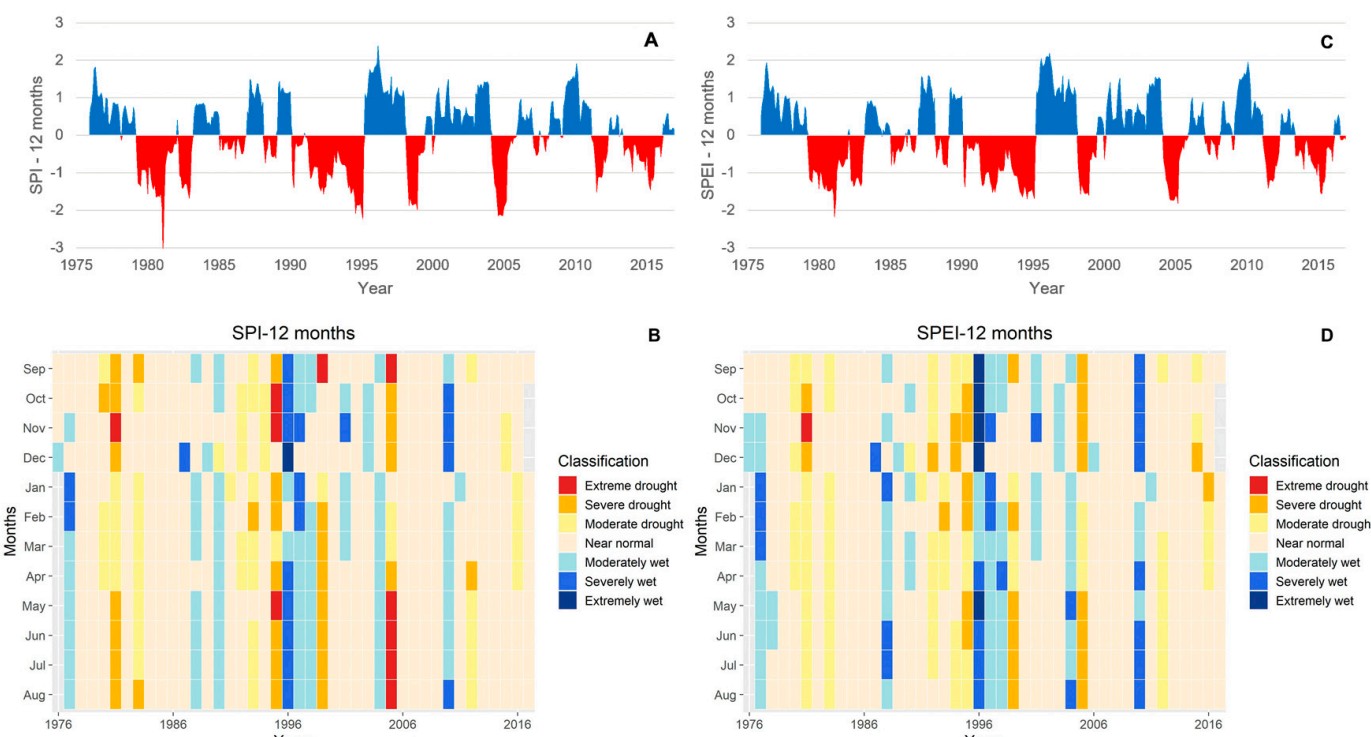

**Figure 2.** Results of SPI (**A**,**B**) and SPEI (**C**,**D**) for 12 months and its classification following values of Figure 1 and (**B**–**D**). (**A**,**C**) Dry values are showed in red and wet values are in blue.

The variation in recharge over time has been examined using the Groundwater Recharge Drought Index (GRDI), as depicted in Figure 3. As outlined in the Methods section, the GRDI was analyzed using two different values of soil moisture capacity for recharge calculations. Figure 3A,B present the results of the GRDI with the soil moisture capacities of 40 mm and 70 mm, respectively. Notably, there is a prolonged five-year drought period (1991–1995) observed in recharge, coinciding with one of the driest periods in the SPI and SPEI indices (refer to Figure 2). Furthermore, three wet periods spanning three years each (1996–1998, 2002–2004, and 2010–2012) can be identified, with the first period (1996–1998) being the most pronounced. Figure 3C illustrates the difference between the GRDI results obtained using the two soil moisture capacity values. Higher differences are observed when the index is negative, indicating drought periods. In Figure 3D,E, distinct disparities between the two soil moisture capacities are evident, with droughts appearing more severe in Figure 3D compared to Figure 3E. Wet periods, instead, are more severe in Figure 3D, where the soil moisture capacity is higher (70 mm).

If we compare the drought periods of GRDI with the SPI and SPEI, identified drought periods are mainly the same if we use the 40 mm soil capacity, which makes sense as it means that the soil has less dampening effect on the groundwater recharge. Periods with a greater quantity of recharge on GRDI are the same as the wettest periods registered with SPI and SPEI indices. However, it is important to note that the relationship between meteorological drought indices and soil moisture is not always straightforward and can vary depending on local factors. Hence, it is advisable to complement drought indices with the direct measurements of soil moisture for a more precise evaluation of soil drought conditions. Incorporating methods such as soil moisture sensors, lysimeters, cosmic-ray neutron sensing, or satellite-based observations can provide valuable data to enhance the accuracy of assessing soil moisture deficits. By integrating both indirect drought indices and direct measurements, a more comprehensive understanding of soil drought dynamics can be achieved, enabling better-informed decisions and management strategies.

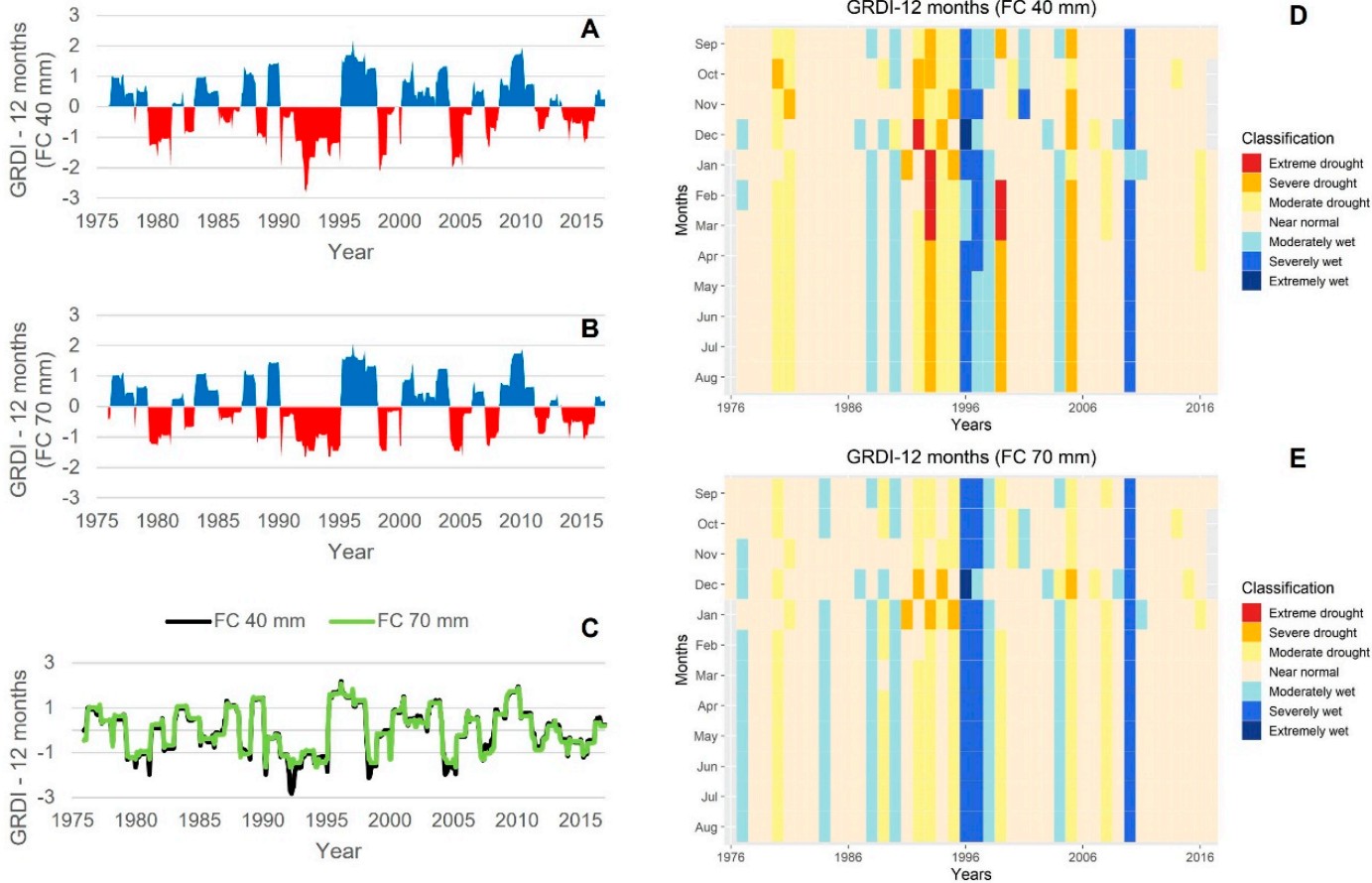

**Figure 3. Figure 3**. (**A**–**E**) Results of GRDI for two different field capacities or FCs of 40 and 70 mm (**A**,**B**). (**C**) shows the differences between both results of GRDI. Classification of GRDI results (**D**,**E**). Blue and red above and below zero, respectively.

### 3.2. Future Projections

Table 3 presents a comparison of basic statistical parameters between the historical modeled data and the future projections. Notably, the average annual precipitation values for the future series are significantly lower, approximately around 80 mm, compared to the historical data. Regarding temperatures, the maximum temperatures exhibit an increase of 1.2 °C relative to historical data in the RPC 4.5 scenario, and a higher increase in 1.6 °C in the RPC 8.5 scenario. Similarly, the minimum temperatures also show an upward trend compared to the historical data, with an increase of 1.1 °C in the RPC 4.5 scenario and a higher increase of 1.4 °C in the RPC 8.5 scenario.

**Table 3.** Statistical comparison between all series data of the study (historical and futures simulations).

| Variable | Data | Max | Min | Mean | Median |
|---|---|---|---|---|---|
| Precipitation (mm) | Historical data (1975–2004) | 1003.95 | 222.10 | 574.69 | 622.06 |
| | Future RCP 4.5 (2030–2059) | 878.27 | 184.17 | 496.08 | 541.17 |
| | Future RCP 8.5 (2030–2059) | 878.10 | 186.54 | 492.74 | 547.11 |
| Maximum temperature (°C) | Historical data (1975–2004) | 25.64 | 22.65 | 23.84 | 23.77 |
| | Future RCP 4.5 (2030–2059) | 26.91 | 23.85 | 25.09 | 25.04 |
| | Future RCP 8.5 (2030–2059) | 27.30 | 23.66 | 25.40 | 25.35 |
| Minimum temperature (°C) | Historical data (1975–2004) | 12.43 | 4.62 | 9.52 | 9.49 |
| | Future RCP 4.5 (2030–2059) | 13.59 | 5.95 | 10.63 | 10.72 |
| | Future RCP 8.5 (2030–2059) | 13.91 | 5.98 | 10.92 | 11.04 |

The results of the indices for the future simulation series (2030–2059) (see Figures 4 and 5) show two long periods of drought of four and five years, one from 2035–2038 and the other from 2046–2050. The years from 2053–2054 were also projected to be two very dry years. The years 2051 and 2052 are the wettest in the studied time-series. The years from 2042 to 2045 will be also humid ones, like the first years of the series 2031 and 2032, and the last, 2059. If we compare the results obtained with the first 29 years of the observed historical data (1975–2004), we have two dry periods, five years, and a dry year. In the case of the future, we also have two longer dry periods. For the recharge, in the observed historical series, there are three very humid years and, on the other hand, for future series, we will only have two.

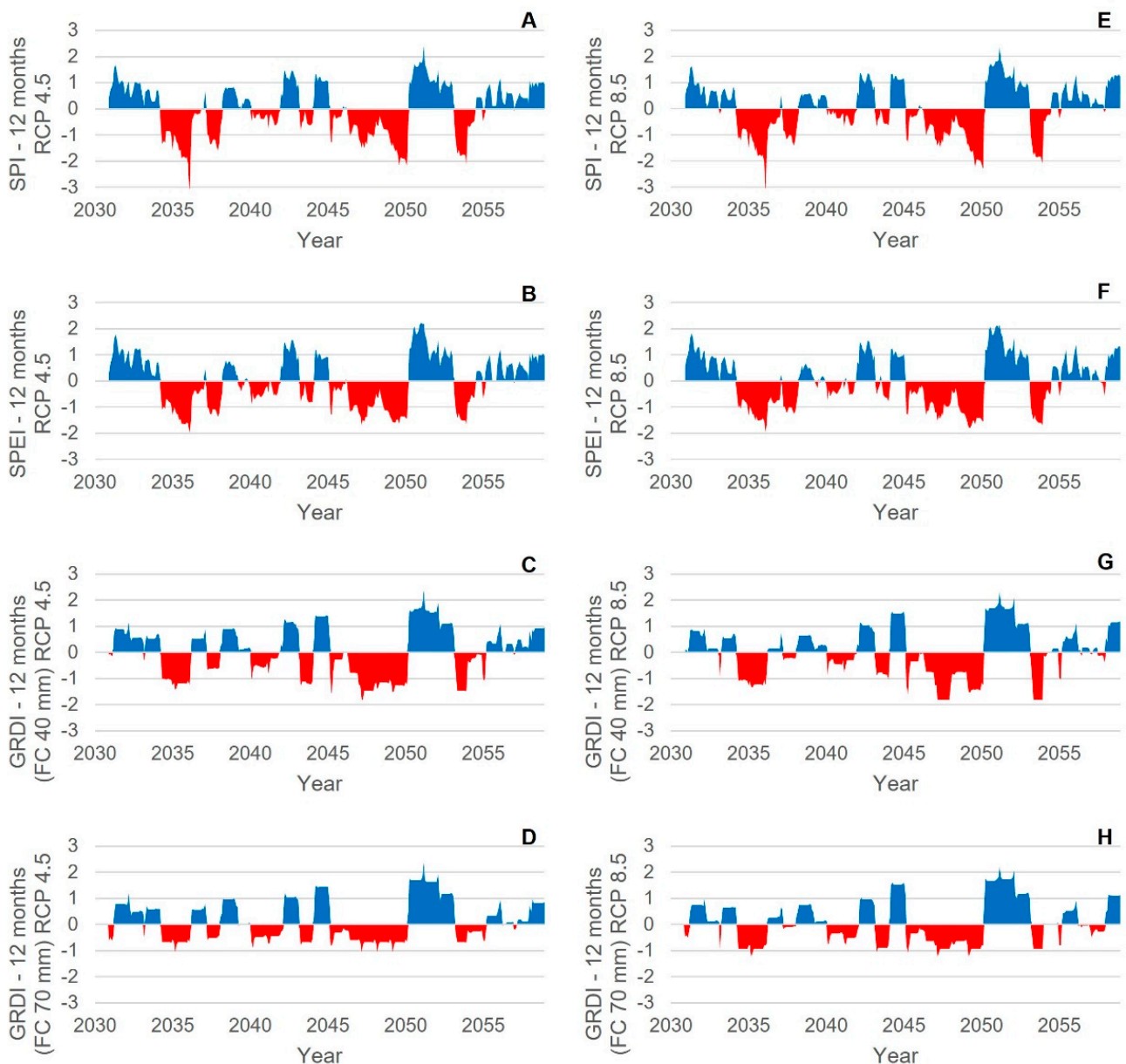

**Figure 4.** (**A**–**H**) Results of the indices SPI, SPEI, and GRDI for future simulations in scenarios RCP 4.5 (**A**–**D**) and RCP 8.5 (**E**–**H**). Blue and red above and below zero, respectively.

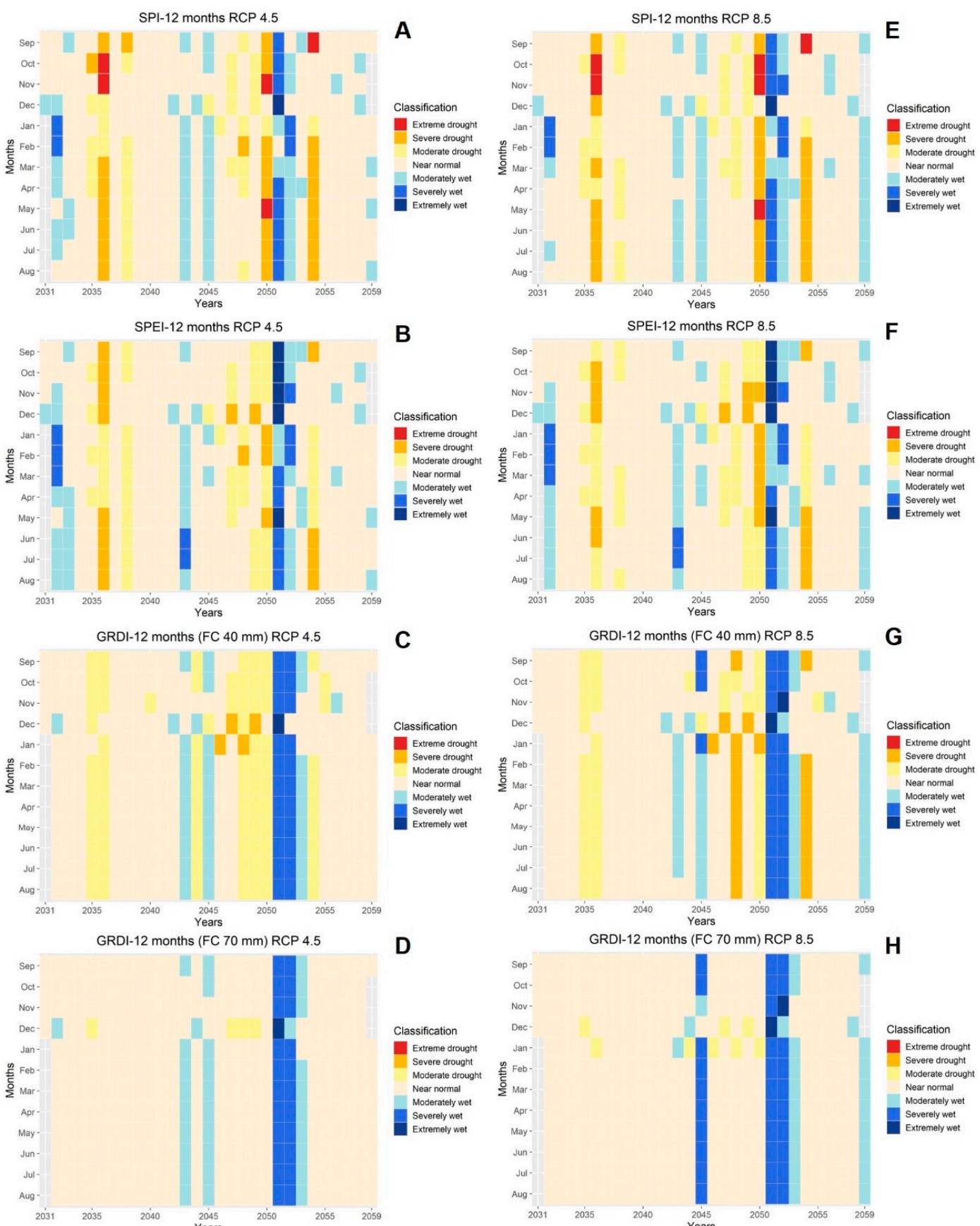

**Figure 5.** (**A**–**H**) Classification of indices for future simulations in scenarios RCP 4.5 (**A**–**D**) and RCP 8.5 (**E**–**H**).

## 4. Discussion

The comparison of the performance of SPI an SPIE in our case study reveals temporal consistency between both indexes ($R^2$: 0.98 between SPI-12 and SPEI-12 during the period 1976–2017). Similar results were obtained in Mongolia [20]. When using SPI or SPEI to estimate the length and intensity of the droughts in the Doñana area, results are very similar. This fact can be interpreted to be of low importance in terms of evaporation in the drought assessment, compared to precipitation.

GRDI estimations with the different soil capacities (40 mm and 70 mm) are very similar for wet periods ($R^2$: 0.96 between GRDI-40 and GRDI-70 during the period 1976–2017). For dry periods, GRDI has more extreme values when the soil capacity is lower, and groundwater recharge is less damped by the soil effect. The length of drought periods remains similar for the three indexes used. Slight differences in GRDI (Figure 5C,D,G,H) are because of the soil parameters that influence recharge.

This study is the first to report a specific future drought classification in the Doñana area, building on the global climate change models. Although there are previous studies that have focused on temperature, precipitation, and groundwater recharge changes [9,18], they do not take into account the occurrences of droughts.

According to the Guadalquivir hydrological planning [21], in the RCP 4.5 scenario (RCP emission stabilization scenario), the average rainfall decrease between 1980–2018 and 2039 will range from 3.0% to 4.6%. We obtained that, for precipitation, there will be a decrease of 13% for the RCP 4.5 and 12% for RCP 8.5. In RCP 8.5 (scenario of increased RCP issuance), the decrease is almost triple, ranging between 10.6% and 13.0%. For the IPCC RCP 4.5 and RCP 8.5 climate scenarios, our results regarding the temperature give similar increase values compared to other studies in the whole Guadalquivir catchment [21]. The average minimum temperature will increase between 11 and 15% and the average maximum temperature between 5% and 6%. Higher values are expected in the Doñana area during the period 2071–2100, with increases between 1.2 °C and 7.4 °C [9].

Consistent with findings from other researchers in the study area, our study also demonstrates that simulated recharge is lower under future climate scenarios. Guardiola-Albert and Jackson [9] estimated that, by the year 2080, aquifer recharge would experience an average decrease of 14–57% compared to historical data from 1975 to 1998. This decrease in recharge is associated with a substantial decline in groundwater levels of up to 17 m. More recent modeling based on the RCP 4.5 scenario for the year 2039 shows an average reduction of 8.5% compared to the average from 1980 to 2018. For the RCP 8.5 scenario, the reduction is more than double, with an average of 18.2% and ranging from 17.6% to 18.8% [21].

In our study, the simulated mean recharge was statistically lower (Steele–Dwass, *p*-value < 0.05) during the middle (2040–2069) to late (2070–2099) 20th century in comparison to the historical period (1950–2009). These results align with the conclusions drawn by Taylor et al. [21] for southern Europe as a whole. Additionally, the similar findings of decreased recharge due to climate scenario simulations have been reported in other regions such as Greece [22] and China [23]. The latest research in Doñana area states that climate change is likely to become the greatest risk to the degradation and availability of water resources in this protected wetland [5]. It is very likely that the greatest threat to many of Doñana's dependent ecosystems, for example, the peridune lagoons, is related to these drought episodes. According to the Guadalquivir Hydrological Plan 2022–2027 [21] there is no likely significant increase in flood risk as a result of climate change. Instead, there will be a higher frequency of droughts, especially 5-year droughts.

Although there was evidence from modeling that climate change could alter the hydrology and ecology of the Doñana in the future, the result of the present study does not pose evidence of an increment of droughts. Our results show a similar frequency of droughts, for the same period length of 30 years: two long periods of drought of four and five years, respectively. The significant difference is obtained for the wet periods. The periods with severely to extremely wet SPIs were 25% lower in future predictions with

respect to the historical time series. In addition, the periods with severely to extremely wet SPEIs were 38% lower in future predictions with respect to the historical time series. This is precisely the current reality, as, since 2011, there has not been any wet year. This phenomenon is resulting water resources scarcity in Doñana area, which also happens in the drought periods. The significant increment in mean and maximum temperatures, which would result in a subsequent increment of ET in the area, reduces recharge rates and adds a negative effect to the OUVs of the park.

In a recent academic report [8], some of the precipitation, temperature and ET data obtained in this research were used to model the daily evolution of the depth of one of the most important OUV ponds of Doñana, namely Santa Olalla pond. This is the only permanent pond of DNP and it is highly dependent on rainfall and the groundwater discharge from a local groundwater system. A one-dimensional model was developed to simulate the daily variations in the depth of the pond from 2030 to 2060. This modeling approach enabled the authors to predict the future condition of the pond. After calibrating and validating the model, the results indicated that the pond would experience complete drying periods starting from 2038. These dry periods would span a total of nine years, accounting for 30% of the modeled time period. Furthermore, the study projected a reduction in rainfall, although maximum daily precipitation events could still reach values as high as 156 mm/day. Another noteworthy finding from this study is the estimation of actual daily direct evaporation from the pond, which was projected to reach 15 mm/day. The average daily evaporation was determined to be 4 mm/day. See "Supplementary Material" for more information about this report [24]: Current water regime and modeling during the 21st century in Santa Olalla pond (Doñana National Park), presented in Spanish in 2021.

In another recent study conducted in Doñana National Park on groundwater recharge in arid and semiarid climates [6] used a spatio-temporal kriging algorithm to analyze the rainfall variability during a long period from 1975 to 2016 and found that monthly recharge estimations range between 21 and 91% of the maximum rainfall, with overestimations in areas with uneven rainfall measurement distribution. The regional average frequency of drought days (FDD) is anticipated to increase substantially in several regions, even during the first half of the 21st century, indicating that the regional drought conditions are likely to shift toward more severe conditions. However, unprecedented regional drought conditions, in which the regional average FDD is larger than the maximum value in the past 145 years and this FDD exceedance lasts longer than five consecutive years, are projected to not be unlikely in some regions by the end of the century, even in a low-emission scenario [4].

*Future Research and Implications for Decision Makers*

Future research in the context of Doñana National Park holds significant implications for decision-makers, particularly concerning global water models (GWMs) and gravimetry (GRACE) telemetry. By incorporating these approaches, together with the advances in the prediction of droughts from the application of SPI, SPEI, and GRDI indices, valuable insights can be gained to enhance water resource management and address potential challenges. Future studies could explore the integration of global water models, which provide a comprehensive understanding of water availability and distribution patterns on a global scale, with local-scale data from Doñana. This integration can enable decision makers to make informed choices regarding water allocation and conservation strategies. Furthermore, the utilization of gravimetry telemetry, such as GRACE, can offer precise measurements of water storage changes, facilitating the monitoring of groundwater levels and the assessment of water availability in the region. These advancements in research hold great potential to guide decision-making processes, ensuring the sustainable management of water resources in Doñana National Park.

Although GWMs are important tools for predicting future drought conditions, they also have inherent model biases. These biases can stem from various factors, such as the inclusion or exclusion of stomatal response to $CO_2$ and vegetation dynamics, domestic and

industrial water withdrawal, and land use. The socioeconomic growth that leads to an increase in water withdrawal can exacerbate stream drought, but it remains challenging to obtain accurate projections or scenarios regarding these sectors. Therefore, developing appropriate and feasible adaptation plans is crucial for overcoming the expected severe dry conditions. Our research highlights potential concerns regarding existing infrastructures and practices that were designed based on historical records or experiences. They may prove insufficient in the near future to cope with droughts and wet periods in a warmer climate, particularly in specific regions. Thus, together with recent geophysical techniques developed to improve knowledge of hydrological variability at the terrestrial scale, such as GRACE [25–27], it is imperative to improve our preparedness in the given time horizon before unprecedented drought conditions arise. Our findings contribute to a better understanding of the factors that influence droughts, decreased wet periods, and the degree of changes in such climatic behavior in DNP in the future.

## 5. Conclusions

The results of this study reveal a significant statistical increase in both maximum and minimum temperatures within the DNP area during the period from 2030 to 2060. The temperature rise ranges from 1.1 °C to 1.6 °C under two different climate change scenarios predicted by the IPCC (RPC 4.5 and RPC 8.5). This temperature increase is expected to contribute to a decline in recharge rates due to increased evapotranspiration. Furthermore, there will be a reduction in the frequency of wet periods, and the mean precipitation is projected to decrease from 575 mm/year to 496 mm/year (RPC 4.5) and 493 mm/year (RPC 8.5) during this period.

The analysis of drought and pluvial events Indicates the occurrence of two significant drought periods and two notable pluvial events. This study suggests that the frequency of future drought periods will be similar to that of the past. However, it is important to note that the future predictions anticipate a decrease in the occurrence of severely to extremely wet periods based on SPI and SPEI indices. These wet periods are projected to be 25% and 38% lower, respectively, compared to the historical time series.

These findings provide valuable information for the competent authority to implement appropriate mitigation measures and establish a robust integrated water resources management (IWRM) framework in the basin. It is crucial for these measures to take into account the anticipated impacts of climate change. Overall, this study highlights the importance of integrating the effects of climate change in the basin's water management strategies.

**Supplementary Materials:** The following supporting information can be downloaded at: https://www.mdpi.com/article/10.3390/w15132369/s1. Faucheux-Vega, M. Current water regime and modeling during the 21st century in Santa Olalla pond (Doñana National Park). Master Thesis Report, University Pablo de Olavide (Seville). 2022, Unpublished.

**Author Contributions:** M.R.-R. and C.G.-A. conceived of the idea for this publication. M.J.M.-V. performed the computations. C.G.-A. and M.J.M.-V. verified the analytical methods. All authors have read and agreed to the published version of the manuscript.

**Funding:** This work has been possible thanks to the collaboration agreement between the Guadalquivir Hydrographic Confederation and the Pablo de Olavide University: "Study of hydrological monitoring and modeling of the pond-aquifer relationship in the Doñana aquifer. Inventory and monitoring" (B.O.E. 23/07/2020, p. 56606). The logistical and technical support of the Singular Scientific-Technical Infrastructure of the Doñana Biological Reserve (DBR-ICTS) is gratefully acknowledged.

**Data Availability Statement:** Data supporting this study are available from ADAPTECCA at https://www.adaptecca.es (accessed on 15 November 2022).

**Conflicts of Interest:** The authors declare no conflict of interest.

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
