# Peer review of "Calculation of the SPI, SPEI, and GRDI Indices for Historical Climatic Data from Doñana National Park: Forecasting Climatic Series (2030–2059) Using Two Climatic Scenarios RCP 4.5 and RCP 8.5 by IPCC"

_water, doi:10.3390/w15132369_

Round 1

Reviewer 1 Report (Previous Reviewer 2)

The manuscript submitted by Vega et al. is a study on SPI and SPEI to predict droughts in a watershed in Spain. In my opinion this study used already known methods to quantify the potential drought index, but the main problem with this paper is that the title is misleading given that no hydrogeological information is provided by the authors about the groundwater resources in the area. In fact, a reader may expect to see groundwater level monitoring or piezometric maps or hydrogeological balance or a hydrogeological conceptual model of the study area. GIven that all these informations are laking I feel that this paper is not suitable for publication in Water.

The level of English is of sufficient quality to allow international readers to understand the meaning of this manuscript.

Author Response

Review of Water 2324522

Comments and suggestions for authors: :

QUESTION: The manuscript submitted by Vega et al. is a study on SPI and SPEI to predict droughts in a watershed in Spain. In my opinion this study used already known methods to quantify the potential drought index, but the main problem with this paper is that the title is misleading given that no hydrogeological information is provided by the authors about the groundwater resources in the area. In fact, a reader may expect to see groundwater level monitoring or piezometric maps or hydrogeological balance or a hydrogeological conceptual model of the study area. Given that all these informations are laking I feel that this paper is not suitable for publication in Water.

Comments on the Quality of English Language: The level of English is of sufficient quality to allow international readers to understand the meaning of this manuscript.

ANSWER: Thank you, we appreciate the comment about the level of English is of sufficient quality to allow international readers to understand the meaning of this manuscript. Regarding the question of the title. We completely agree with the reviewer point o view. Reviewer 2 has the same concern. So, to avoid confusion, and given that no hydrogeological information is provided by the authors about the groundwater resources in the area, groundwater level monitoring, piezometric maps or hydrogeological balance, we have decided to change the title of the MS to: “Calculation of the SPI, SPEI, and GRDI indices for historical climatic data from Doñana National Park. Forecasting climatic series (2030-2059) using two climatic scenarios RCP 4.5 and RCP 8.5 by IPCC.”

Specific comments

No specific comments are included in the revision

We acknowledge the insight of the reviewer and are convinced that, together with the extensive Specific comments of reviewer 2 and reviewer 3, the new version of our work is much improved and suitable for publication in Water.

Reviewer 2 Report (New Reviewer)

In the reviewer’s opinion, the submitted manuscript may hold the potential to be suitable for publication in Water subject to several recommended modifications.

The motivation of the paper under review is unclear. According to the title, the paper should focus on determining the characteristic of groundwater resources in the context of Global Change.

However, according to the authors, the main objectives of this paper are (line 82):

1. Calculation of the SPI, SPEI, and GRDI indices for historical climatic data from DNP.

2. Comparing these indices with those applied by the managers of DNP.

3. Applying these indices to forecast climatic series (2030-2059) using two climatic scenarios RCP 4.5 and RCP 8.5 by IPCC.

SPI based on precipitation and SPEI based on precipitation and air temperature have nothing to do with groundwater. Indices applied by the managers of DNP are unknown to the readers. Therefore the title is not in line with the abstract and introduction.

Detailed comments

Line 39

Define  OUV - please refer to, e.g. Operational Guidelines for the Implementation of the World Heritage Convention (WHC.19/01 - 10 July 2019) or define “To be deemed of Outstanding Universal Value, a property must also meet the conditions of integrity and/or authenticity and must have an adequate protection and management system to ensure its safeguarding.”

Line 45

It remains difficult to determine the onset, intensity, and duration of drought in spite of a great number of studies committed to this effort [3].

To support this statement the authors should not refer to a 20-year-old paper.

Line 148

Why is the selected timescale only SPI-12? Is it because other scales are not compatible with the CHG approach? Because is often used?

When SPI is computed for medium accumulation periods (e.g., 3 to 12 months), it can be used as an indicator for reduced stream flow and reservoir storage.

When SPI is computed for longer accumulation periods (e.g., 12 to 48 months), it can be used as an indicator for reduced groundwater recharge.

The reviewer recommends performing calculations for other time scales.

Line 172

… the SPEI can be also calculated at 6 months (SPEI-6) and 12 months (SPEI-12)

Line 173

12 months (SPEI-12) months [2].

Line 187-189

For the present GRDI computations, the recharge time series was calculated using a soil water balance 188 computing program [16].

As the reference [16] is in Spanish (useless for the majority of readers) and the code, TRASERO V2.3 is not available, please describe the technique of calculation of GRDI! In particular please plot selected distribution functions.

Line 197

Why were selected field capacities of 40 and 70 mm?

Line 331

The comparison of the performance of SPI and SPIE in our study case reveals temporal consistency between both indexes, though differences appeared at short timescales.

The short time scales are not discussed in the paper!!!!

Line 340

The small-scale differences are because of the soil parameters that influence recharge.

The short time scales are not discussed in the paper!!!!

Line 381

two long two long

lines 346-432

These sections can be published as supplementary material or partially used in the introduction.

These sections have nothing to do with groundwater.

Last but not least, is there any correlation (expressed numerically) between the analyzed indices? Is it possible to conclude soil drought based on SPI or SPEI?

Too many references in Spanish.

Author Response

Review of Water 2324522. June 2023. Seville.

Comments and Suggestions for Authors

QUESTION: In the reviewer’s opinion, the submitted manuscript may hold the potential to be suitable for publication in Water subject to several recommended modifications. The motivation of the paper under review is unclear. According to the title, the paper should focus on determining the characteristic of groundwater resources in the context of Global Change.

However, according to the authors, the main objectives of this paper are (line 82):

  1. Calculation of the SPI, SPEI, and GRDI indices for historical climatic data from DNP.
  2. Comparing these indices with those applied by the managers of DNP.
  3. Applying these indices to forecast climatic series (2030-2059) using two climatic scenarios RCP 4.5 and RCP 8.5 by IPCC. SPI based on precipitation and SPEI based on precipitation and air temperature have nothing to do with groundwater. Indices applied by the managers of DNP are unknown to the readers. Therefore the title is not in line with the abstract and introduction.

ANSWER: Thank you, we appreciate the comment about the suitability of the paper to be published in Water, and are thankful for it. First of all, we completely agree with the issue of the Title. In that sense, we have decided to completely change the title of this work. The new title goes for: “Calculation of the SPI, SPEI, and GRDI indices for historical climatic data from Doñana National Park. Forecasting climatic series (2030-2059) using two climatic scenarios RCP 4.5 and RCP 8.5 by IPCC.”

Detailed comments

QUESTION: Line 39 Define OUV - please refer to, e.g. Operational Guidelines for the Implementation of the World Heritage Convention (WHC.19/01 - 10 July 2019) or define “To be deemed of Outstanding Universal Value, a property must also meet the conditions of integrity and/or authenticity and must have an adequate protection and management system to ensure its safeguarding.”

ANSWER: Thanks. In the new version of the MS we have modified the text and we have included a new paragraph as suggested by the reviewer: “To be deemed of Outstanding Universal Value, a property must also meet the conditions of integrity and/or authenticity and must have an adequate protection and management system to ensure its safeguarding, as stated in the Operational Guidelines for the Implementation of the World Heritage Convention.”

QUESTION: Line 45. It remains difficult to determine the onset, intensity, and duration of drought in spite of a great number of studies committed to this effort [3]. To support this statement the authors should not refer to a 20-year-old paper.

ANSWER: Thanks. We have decided to include reference number 4, “The relationship of drought frequency and duration to time scales” which is indeed related to the onset, intensity, and duration of drought, and also reference number 2, “A Multiscalar Drought Index Sensitive to Global Warming: The Standardized Precipitation Evapotranspiration Index” to be more accurate, as the referee points out.

QUESTION: Line 148. Why is the selected timescale only SPI-12? Is it because other scales are not compatible with the CHG approach? Because is often used? When SPI is computed for medium accumulation periods (e.g., 3 to 12 months), it can be used as an indicator for reduced stream flow and reservoir storage. When SPI is computed for longer accumulation periods (e.g., 12 to 48 months), it can be used as an indicator for reduced groundwater recharge. The reviewer recommends performing calculations for other time scales.

ANSWER: Correct, we agree with your question. When SPI is computed for longer accumulation periods (e.g., 12 which is the case of this study), it is often used as an indicator for reduced groundwater recharge. We performed calculations for other time scales (in particular, SPI-6 and SPEI-6) and noticed that results were poor to be related to reduced groundwater recharge, so that is the reason why we decided to only maintain the results of SPI-12 for the purpose of this particular study. If the reviewer is interested, we will be pleased to give him/her all the results obtained for SPI-6 and SPEI-6. In any case, we have made changes in the paragraph accordingly. We have stated that we made calculations for SPI-6 in a previous step of this investigation: “In any case, as a previous step of this investigation, we calculated SPI-6 (6 months) and analyze the results obtained. This time-step is better suited as an indicator for reduced stream flow and reservoir storage.”

QUESTION: Line 172… the SPEI can be also calculated at 6 months (SPEI-6) and 12 months (SPEI-12). Line 173. 12 months (SPEI-12) months [2].

ANSWER: Thanks. In the new version of the MS we have corrected that typo: …and 12 months (SPEI-12).

QUESTION: Line 187-189. For the present GRDI computations, the recharge time series was calculated using a soil water balance computing program. As the reference is in Spanish (useless for the majority of readers) and the code, TRASERO V2.3 is not available, please describe the technique of calculation of GRDI! In particular please plot selected distribution functions.

ANSWER: Thank you. In fact, the code TRASERO V2.3 is indeed available and free-to-use through this link:

https://ciclohidrico.com/download/tratamiento-y-gestion-de-series-temporales-hidrologicas/

Although this is a good example of “grey” literature, i.e., academic reports or technical information that it is normally written in Spanish and that it is difficult or nearly impossible to read elsewhere, in this particular case the code is available. Although it is true that it is written in Spanish, it is used in many countries due to its friendly interface and because it is quite simple to use. The reference: “A. Padilla Benítez y J. Delgado Pastor, «Trasero 2.3. Tratamiento y gestión de series temporales hidrológicas» Diputación de Alicante. Ciclo Hídrico, Alicante, 2015” is also available via the installation package. In any case, in the new version of the MS, we have included a new paragraph in which a technique of calculation for effective rainfall is explained: “The Thornthwaite method is a widely used approach for estimating effective rainfall, which refers to the amount of precipitation available for soil moisture recharge and plant growth. At a daily scale, the effective rainfall calculation through the Thornthwaite method involves several steps. Firstly, the potential evapotranspiration (PET) is estimated using the Hargreaves equation, which takes into account maximum and minimum daily temperature data and the latitude of the location. The PET represents the amount of water that would evaporate and transpire under optimal moisture conditions. Next, the field capacity (i.e., maximum soil moisture for the site) is needed as a site-specific parameter. The field capacity, in mm, helps determine the water deficit or surplus for a given day. By summing the daily rainfall values, an estimate of the actual evapotranspiration (AET) is obtained. The effective rainfall, also in mm, can then be determined by subtracting the AET from the total rainfall. This approach provides valuable insights into the water availability and helps in managing irrigation and water resources at a daily scale.

QUESTION: Line 197. Why were selected field capacities of 40 and 70 mm?

ANSWER: Thanks. The recharge is very sensible to the field capacity values selected. In this type of studies, it is very common to use an upper and a lower boundary to cope with uncertainty. Anyhow, for the type of soil in Doñana’s dunes, a value of 40 to 70 mm has been used to calibrate groundwater models to assess the water management for Doñana groundwater bodies, so this the reason to pick such values and implement them to this particular investigation.

QUESTION: Line 331. “The comparison of the performance of SPI and SPIE in our study case reveals temporal consistency between both indexes, though differences appeared at short timescales”. The short time scales are not discussed in the paper!!!!

ANSWER: Thanks. As the reviewer suggests, we have decided to eliminate this phrase, to be consistent. In fact, short time-scales are not discussed in the paper, although SPEI-6 and SPI-6 were calculated in a previous step, as mentioned before.

QUESTION: Line 340. The small-scale differences are because of the soil parameters that influence recharge. The short time scales are not discussed in the paper!!!!

ANSWER: Thank you, we appreciate the comment about the short time scale. In this occasion, the authors are not talking about SPI-12 vs. SPI-6, but about the slight differences in extremely wet, severely wet or moderately wet periods, etc., found when using GRDI 40 mm (Figs. 5C, 5G) vs. GRDI 70 mm (Figs. 5D, 5H). In that sense, we have decided to rephrase the sentence in Line340 in order to clarify. In the new version of the MS, the phrase goes like this: “Slight differences in GRDI (Fig. 5C, 5G and Fig. 5D, 5H) are because of the soil parameters that influence recharge.”

QUESTION: Line 381. two long two long.

ANSWER: As the reviewer correctly points out, we have eliminated the repeated phrase “two long” in the new version of the MS.

QUESTION: lines 346-432. These sections can be published as supplementary material or partially used in the introduction. These sections have nothing to do with groundwater.

ANSWER: Thanks. According to the suggestions made by the reviewer, in the new version of the MS we have move some of the paragraphs to the “Introduction” section: “Moreover, the information that we have complied from EUROCODEX project for the Doñana area is quite in agreement with the studies performed over all the Guadalquivir catchment for climate change scenarios (i.e., RCP 4.5 or RCP 8.5 scenario from IPCC panel).” Also, we have relocated some in the “Discussion” sections. Those are the ones referring to our actual findings: “We have obtained that for precipitation there will be a decrease of a 13% for the RCP 4.5 and 12% for RCP 8.5. In RCP 8.5 (scenario of increased RCP issuance) the decrease is almost triple, ranging between 10.6% and 13.0%. For the IPCC RCP 4.5 and RCP 8.5 climate scenarios, our results about the temperature give similar values of increase than other studies in the whole Guadalquivir catchment [21]. The average minimum temperature will increase between 11 and 15% and the average maximum temperature between 5% and 6%. Higher values are expected in the Doñana area for the period 2071-2100, with increases between 1.2°C and 7.4°C [9].

QUESTION: Last but not least, is there any correlation (expressed numerically) between the analyzed indices? Is it possible to conclude soil drought based on SPI or SPEI?

ANSWER: Thanks. Regarding correlation expressed numerically between SPI, SPEI and GRDI, such computations were already made and there is a high correlation (R2=0.97) between SPI-12 and SPEI-12 between the period 1976-2017 (see figure):

Also, there is a high correlation between GRDI-40 and GRDI-70 between the period 1976 and the period 2017:

These findings have been made explicit in the new version of the MS (L510 and L518, respectively).

As for the second question. To infer soil drought from these indices, it is important to understand the relationship between meteorological drought and soil moisture content. Soil moisture is influenced by various factors, including precipitation, evapotranspiration, soil properties, and vegetation cover. However, meteorological drought can serve as an early indicator of potential soil moisture deficits. By examining the SPI or SPEI values and considering the local soil characteristics and vegetation conditions, it is possible to stablish strong relations and conclusions about soil drought conditions. In Doñana region, after experiencing a prolonged period of negative SPI or SPEI values, it suggests that there is a higher likelihood of soil moisture deficit in the sand dunes (arenosols). However, it is important to note that the relationship between meteorological drought indices and soil moisture is not always straightforward and can vary depending on local factors.

“Therefore, it is recommended to combine drought indices with direct measurements of soil moisture, such as using soil moisture sensors or satellite-based observations, to obtain a more accurate assessment of soil drought conditions.”

We have included the paragraph in italics in the new version of the MS (L338-342)

QUESTION: Too many references in Spanish for the entire manuscript. The rest of the manuscript also is in need of a thorough edit.

ANSWER: Thank you, we appreciate the comment about the references in Spanish, known also as “grey” literature. We are convinced that such “grey” literature written in Spanish is important for understanding scientific advances in hydrological sites such as Doñana in southern Spain (and other locations in Latin America). It provides localized information, rapid communication of research findings, multidisciplinary perspectives, data availability, and access to Spanish-language research that may not be available through traditional academic channels. By considering grey literature, researchers can gain a more comprehensive understanding of the hydrology, ecology, and socio-economic aspects of these sites, facilitating informed decision-making and sustainable management practices. For example, such information is vital for establishing a baseline to measure impacts in DNP. Significant abstraction for agriculture began in the 1973-74 (Manzano 2001), so its impacts have been reported in grey literature for decades (Bea et al. 2021), and the Spanish media has reflected concerns among scientists and managers about impacts on water quantity and quality since the late 1970s (https://elpais.com/diario/1979/11/17/espana/311641209_850215.html).

  • Manzano, M. 2001. Los humedales de Doñana y su relación con el agua subterránea. Pp. 161-167 in: Reunion internacional de expertos sobre la regeneración hídrica de Doñana. Ministerio de Medio Ambiente.
  • Bea Martínez, M.; Fernández Lop, A.; Gil, T.; Seiz Puyuelo, R. y cols. (2021). El robo del agua. Cuatro ejemplos flagrantes del saqueo hídrico en España. WWF España.

Anyhow, three new and up-to-date references published in international Journals has been included in the last section of the MS (new references 24-25-26):

  • [24] Mohamed, A., Faye, C., Othman, A., & Abdelrady, A. (2022). Hydro-geophysical Evaluation of the Regional Variability of Senegal’s Terrestrial Water Storage Using Time-Variable Gravity Data. Remote Sensing, 14(16), 4059.
  • [25] Othman, A., Abdelrady, A., & Mohamed, A. (2022). Monitoring Mass Variations in Iraq Using Time-Variable Gravity Data. Remote Sensing, 14(14), 3346.
  • [26] Vishwakarma, B. D. (2020). Monitoring droughts from GRACE. Frontiers in Environmental Science, 8, 584690.

As for the second question, a thorough edit has been applied in the new version of the MS:

  • The Abstract was rewritten and in the new version of the MS goes for: “In this study, we utilized three different indices to assess drought conditions in the Doñana National Park (DNP) located in southern Spain. These indices included the Standardized Precipitation Index (SPI), which is based on precipitation statistics, the Standardized Precipitation Evapotranspiration Index (SPEI), which incorporates both precipitation and air temperature data, and the Groundwater Recharge Drought Index (GRDI), a newly developed index specifically designed to evaluate groundwater drought. The analysis covered the time period from 1985 to 2015, and future projections were made for the years 2030 to 2060 under different climate scenarios (RCP 4.5 and RCP 8.5). Our findings revealed a significant decrease in total precipitation of approximately 13-14% compared to historical records (1985-2015). Moreover, severely to extremely wet periods exhibited a reduction ranging from 25% to 38%. A key contribution of this study is the application of the GRDI index, which allowed us to assess groundwater recharge rates. We ob-served a decline in simulated mean recharge rates during the 21st century when compared to the historical period spanning from 1950 to 2009. This decline can be attributed to increased evapotranspiration. The results of this research provide valuable insights for the Spanish water re-sources administration. The observed reductions in precipitation and groundwater recharge rates emphasize the need for appropriate mitigation measures. The findings will aid the administration in formulating an integrated water resources management strategy in the Doñana National Park and its surrounding basin. By understanding the projected changes in drought conditions, the administration can make informed decisions to ensure sustainable water resource management in the region.

Other significant changes are (according to line numeration of the new version of the MS):

  • L93-98: “One of the main significant challenges in drought studies is the limited availability of long-term climatic datasets, which are crucial for the development of robust climatic models, including drought predictive models. Over the past few decades, numerous efforts have been made to develop and enhance drought indices. Among these, the standardized precipitation index (SPI) has emerged as a valuable tool for assessing drought dynamics in a specific region”
  • L299-302: “The historical time series of observations spanned from October 1st, 1975 to September 30th, 2017, following the hydrological year. To facilitate the analysis, the daily data were aggregated into monthly values. The R programming language, specifically the SPEI package, was utilized for further analysis and calculations
  • L316-325: “Daily precipitation, maximum and minimum temperature data from the years 2030 to 2059 have been acquired for the upcoming climate series. This data was obtained from the Spanish Climate Change Office's platform on adaptation to climate change (http://adaptecca.es/en), which sources its data from the Spanish State Agency of Meteorology [17]. These data have undergone small-scale dynamic downscaled processing. This approach is preferred for our study as it provides a physically consistent scaling between variables, such as rainfall and temperature, unlike statistical methods. The data was sourced from the EUROCORDEX project [18], which is a comprehensive initiative aimed at generating high-resolution climate projections specifically tailored to the European region.
  • L376-387: “Figure 2 illustrates the temporal progression of the Standardized Precipitation Index (SPI) and the Standardized Precipitation-Evapotranspiration Index (SPEI). Notably, three prolonged drought periods consisting of four to five consecutive dry years can be observed: 1980-1984, 1991-1995, and 2014-2017. Additionally, there are two instances of two consecutive dry years: 2005-2006, along with two remarkably dry years in 1999 and 2012. The SPI and SPEI indices exhibit similar patterns, although the SPEI tends to display less extreme values compared to the SPI. It appears that dry periods are relatively mitigated in comparison to wet periods. This phenomenon can be attributed to the consideration of evapotranspiration in the SPEI calculation, which has a greater impact on dry periods. Shifting focus to wet periods, there are two four-year periods (1987-1990 and 2001-2004), one three-year period (1996-1998), and two biennial periods (1976-1977 and 2010-2011).
  • L417-429: “The variation in recharge over time has been examined using the Groundwater Recharge Drought Index (GRDI), as depicted in Figure 3. As outlined in the Methods section, the GRDI was analyzed using two different values of soil moisture capacity for recharge calculations. Figures 3A and 3B present the results of the GRDI with soil moisture capacities of 40 mm and 70 mm, respectively. Notably, there is a prolonged five-year drought period (1991-1995) observed in recharge, coinciding with one of the driest periods in the SPI and SPEI indices (refer to Fig. 2). Furthermore, three wet periods spanning three years each (1996-1998, 2002-2004, and 2010-2012) can be identified, with the first period (1996-1998) being the most pronounced. Figure 3C illustrates the difference between the GRDI results obtained using the two soil moisture capacity values. Higher differences are observed when the index is negative, indicating drought periods. In Figures 3D and 3E, distinct disparities between the two soil moisture capacities are evident, with droughts appearing more severe in Figure 3D compared to Figure 3E.
  • L468-475: “Table 3 presents a comparison of basic statistical parameters between the historical modeled data and the future projections. Notably, the average annual precipitation values for the future series are significantly lower, approximately around 80 mm, compared to the historical data. Regarding temperatures, the maximum temperatures exhibit an increase of 1.2 ºC relative to the historical data in the RPC 4.5 scenario, and a higher increase of 1.6 ºC in the RPC 8.5 scenario. Similarly, the minimum temperatures also show an upward trend compared to the historical data, with an increase of 1.1 ºC in the RPC 4.5 scenario and a higher increase of 1.4 ºC in the RPC 8.5 scenario.
  • L544-549: “Consistent with findings from other researchers in the study area, our study also demonstrates that simulated recharge is lower under future climate scenarios. Guardiola-Albert and Jackson [9] estimated that by the year 2080, aquifer recharge would experience an average decrease of 14-57% compared to historical data from 1975-1998. This decrease in recharge is associated with a substantial decline in groundwater levels of up to 17 meters. More recent modeling based on the RCP 4.5 scenario for the year 2039 shows an average reduction of 8.5% compared to the average from 1980-2018. For the RCP 8.5 scenario, the reduction is more than double, with an average of 18.2% and ranging from 17.6% to 18.8% [21].”
  • L656-666: A one-dimensional model was developed to simulate the daily variations in the depth of the pond from 2030 to 2060. This modeling approach enabled the authors to predict the future condition of the pond. After calibrating and validating the model, the results indicated that the pond would experience complete drying periods starting from 2038. These dry periods would span a total of nine years, accounting for 30% of the modeled time period. Furthermore, the study projected a reduction in rainfall, although maximum daily precipitation events could still reach values as high as 156 mm/day. Another noteworthy finding from this study is the estimation of actual daily direct evaporation from the pond, which was projected to reach 15 mm/day. The average daily evaporation was determined to be 4 mm/day.
  • L714-731: The results of this study reveal a significant statistical increase in both maximum and minimum temperatures within the DNP area during the period from 2030 to 2060. The temperature rise ranges from 1.1 °C to 1.6 °C under two different climate change scenarios predicted by the IPCC (RPC 4.5 and RPC 8.5). This temperature increase is expected to contribute to a decline in recharge rates due to increased evapotranspiration. Furthermore, there will be a reduction in the frequency of wet periods, and the mean precipitation is projected to decrease from 575 mm/year to 496 mm/year (RPC 4.5) and 493 mm/year (RPC 8.5) during this period.The analysis of drought and pluvial events indicates the occurrence of two significant drought periods and two notable pluvial events. The study suggests that the frequency of future drought periods will be similar to that of the past. However, it is important to note that the future predictions anticipate a decrease in the occurrence of severely to extremely wet periods based on SPI and SPEI indices. These wet periods are projected to be 25% and 38% lower, respectively, compared to the historical time series.These findings provide valuable information for the competent authority to implement appropriate mitigation measures and establish a robust Integrated Water Resources Management (IWRM) framework in the basin. It is crucial for these measures to take into account the anticipated impacts of climate change.”

Reviewer 3 Report (New Reviewer)

I have posted the comments on the manuscript.

Best Wishes

Author Response

Thanks. Answers to Reviewer 3 are embedded in the pdf file attached. 

Given the importance of this issue, we hope that you will give our manuscript full attention, and that subsequently our paper will be accepted in Water, and will become a valuable reference for future authors interested in Mediterranean wetlands and in water management.

Round 2

Reviewer 1 Report (Previous Reviewer 2)

I am positively impressed by the changes that the authors have provided in their revised manuscript, I feel that since the ambiguity on the title and conclusions on groundwater resources have now been eliminated the paper is ready to be acepted as a valuable contribution to help predicting local scale impact of climate changes to water managers in semiarid regions.

Author Response

Review of Water 2324522. June 2023. Seville.

ROUND 2

REVIEWER 1

Reviewer 1: I am positively impressed by the changes that the authors have provided in their revised manuscript, I feel that since the ambiguity on the title and conclusions on groundwater resources have now been eliminated the paper is ready to be accepted as a valuable contribution to help predicting local scale impact of climate changes to water managers in semiarid regions.

ANSWER: Thank you, we appreciate the comment about the suitability of the paper to be accepted as a valuable contribution to help predicting local scale impact of climate changes to water managers. We are very thankful for it.

Reviewer 2 Report (New Reviewer)

No answer to the questions:

Why is the selected time scale only SPI-12?

Why were chosen field capacities of 40 and 70 mm?

Is there any correlation between the analyzed indices?

Is it possible to conclude soil drought based on SPI or SPEI?

Author Response

Review of Water 2324522. June 2023. Seville.

ROUND 2

REVIEWER 2

Comments and Suggestions for Authors

QUESTION 1: Why is the selected time scale only SPI-12?

ANSWER: When SPI is computed for a 12-month period, which is the case of this study, it is often used as an indicator for reduced groundwater recharge. During the first stages of our investigation, we also performed calculations for other time scales (in particular, SPI-6, but also SPEI-6) and noticed that results were poor to be related to reduced groundwater recharge. That was the reason why we decided to focus on the results of SPI-12, for the purpose of this particular study. If the reviewer is interested, we will be pleased to give him/her all the results obtained for SPI-6 and SPEI-6. In any case, we have made changes in the paragraph accordingly. We have stated that we made calculations for SPI-6 in a previous step of this investigation: “In any case, as a previous step of this investigation, we calculated SPI-6 (6 months) and analyze the results obtained. This time-step is better suited as an indicator for reduced stream flow and reservoir storage. Moreover, this is the time scale chosen by CHG to manage droughts in the drought management plan of the Doñana area. Therefore, three of the great strengths of the SPI drought index are its simplicity, flexibility, and temporal versatility.”

QUESTION 2: Why were chosen field capacities of 40 and 70 mm?

ANSWER: Thanks. The sensitivity of recharge greatly depends on the chosen field capacity values. In studies of this nature, it is common practice to establish upper and lower boundaries to account for uncertainties. However, for the specific soil type found in Doñana's dunes, a range of 40 to 70 mm has been employed to calibrate groundwater models for assessing water management in the Doñana groundwater bodies and in several subsequent papers (e.g., Naranjo-Fernández et al, 2020). This is the rationale behind selecting and applying these particular values to the present investigation. In the new version of the MS, we have included a new paragraph to explain why where chosen field capacities 40 mm and 70 mm

(L225-229): The runoff threshold was assigned based on tabulated values that depend on soil type, vegetation cover, hydrological characteristics and land uses. We have used the same ranges as the ones used in Naranjo-Fernández et al. (2020) for the aeolian sand zone, in which the runoff threshold for each sub-area was estimated and then the direct runoff values were determined [6]”.

  1. Naranjo-Fernández, C. Guardiola-Albert, H. Aguilera, C. Serrano-Hidalgo, M. Rodríguez-Rodríguez, A. Fernández-Ayuso, F. Ruíz-Bermudo, E. Montero (2020) Relevance of spatio-temporal rainfall variability regarding groundwater management challenges under global change: case study in Doñana (SW Spain). Stochastic Environmental Research and Risk Assessment (Q1) DOI: 10.1007/s00477-020-01771-7

* Reference number 6 “[6]” in this paper

QUESTION 3: Is there any correlation between the analyzed indices?

ANSWER: As the reviewer suggest, computations to correlate the analyzed indices were already made. It is noteworthy that a high correlation (R2=0.97) has been observed between SPI-12 and SPEI-12 during the period 1976-2017, as determined from the previous computations. This indicates a strong association between the Standardized Precipitation Index (SPI) and the Standardized Precipitation Evapotranspiration Index (SPEI) over the specified timeframe. Also, there is a high correlation between GRDI-40 and GRDI-70 between the period 1976 and the period 2017.

QUESTION 4: Is it possible to conclude soil drought based on SPI or SPEI?

ANSWER: In order to infer soil drought using the SPI and SPEI indices, it is crucial to comprehend the connection between meteorological drought and soil moisture content. Soil moisture is influenced by multiple factors, such as precipitation, evapotranspiration, soil properties, and vegetation cover. However, meteorological drought can act as an early indicator of potential soil moisture deficits. By analyzing the SPI or SPEI values and taking into account local soil characteristics and vegetation conditions, it is possible to establish strong relationships and draw conclusions regarding soil drought conditions. In the Doñana region, a prolonged period of negative SPI or SPEI values suggests a higher probability of soil moisture deficit in the sand dunes (arenosols). Nonetheless, it is important to acknowledge that the relationship between meteorological drought indices and soil moisture is not always straightforward and can vary depending on local factors.

There is a paragraph in the new version of the MS regarding this question:

L342-245: “Hence, it is advisable to complement drought indices with direct measurements of soil moisture for a more precise evaluation of soil drought conditions. Incorporating methods like soil moisture sensors, lysimeters, cosmic-ray neutron sensing, or satellite-based observations can provide valuable data to enhance the accuracy of assessing soil moisture deficits. By integrating both indirect drought indices and direct measurements, a more comprehensive understanding of soil drought dynamics can be achieved, enabling better-informed decisions and management strategies.”

We acknowledge the insight of the reviewer and are convinced that the new version of our work is improved and suitable for publication in Water.

This manuscript is a resubmission of an earlier submission. The following is a list of the peer review reports and author responses from that submission.

Round 1

Reviewer 1 Report

Review of Water 2324522

Overall comments:

This manuscript has the potential to be an important contribution to the literature, but it is poorly presented and difficult to understand.  I was tempted to simply recommend it be rejected, but the main conclusions are important, both for scientists and resource managers.  Therefore, I think it can be substantially modified to make it suitable for publication.

The manuscript gets off to a disappointing start with an abstract that is not useful at all.  It tells the reader very little about the study or even what was found.  All we know is that precipitation was reduced during the second half of the study.  But what did that do to groundwater resources?  The reader learns that recharge rates are reduced, of course.  By how much?  That is what the reader really needs to know.  Based on the title, the paper is about groundwater and drought.  But where is drought mentioned in the abstract?  It isn’t.  You mention three indicators, but you don’t say what the indicators are related to.  You then talk about Outstanding Universal Value ecosystems, assuming that readers know what this is.  I do not.  Later, in the introduction, I find these are ponds and lagoons.  You state in the abstract that these ecosystems are dependent on groundwater recharge.  Knowing these are ponds and lagoons, I think you actually mean that they are dependent on groundwater discharge, not recharge.  For these many reasons, the abstract is very confusing and needs heavy revision.

The rest of the manuscript is similarly confusing due to poor organization and presentation.  I provide numerous suggestions for improvement in the Introduction section, as an indication of what I mean by poor paragraph structure and presentation.  However, I did not do that for the entire manuscript.  The rest of the manuscript also is in need of a thorough edit.

Specific comments

Abstract:

The abstract should stand alone.  Please spell out SPI, SPEI and GRDI, or at least explain what these indexes relate to.  I looked up GRDI and I still don’t know what it is.  I doubt it is a Global Retail Development Index, nor is it likely a Gridded Relative Deprivation index.  Just reading the abstract thus far, I have no idea what your manuscript is about.

You write, “results are consistent with previous studies” but you don’t say what results or what the previous studies pertain to. 

You state that the most important contribution of your work is the GRDI index, but you never say what that index is, or what it is used for.  Evidently it is related to mitigation measures, but mitigation for what?  This entire abstract is very frustrating to read and should be completely revised.  It tells the reader almost nothing.

Introduction:

29   I know what a method is, or an approach.  But what is a methodological approach?  I suggest replacing “methodological approach” with “multiple approaches.” 

31   I suggest replacing “acknowledged” with “quantified”.  Also, revise to write “. . . to allow a better understanding of the relative importance of specific hydrological processes.” 

34-35   “the benefit of using groundwater is evident” is not useful.  Why is it evident?  I suggest replacing with “Densely populated or agricultural coastal areas, where access to fresh surface water commonly is limited, are largely dependent on groundwater.” 

36   You need to provide a citation for your 80 percent value.

38   What is an outstanding universal value pond or lagoon?  You need to indicate which organization creates or uses this distinction, how it is determined, and cite a source for this term.

40   Safe alternative to what?  I suggest “as a safe alternative to surface water”

42   Beginning a sentence with “on the other hand” implies that this sentence is presenting an alternate perspective relative to the previous sentence.  It is not.  It is just providing additional information.  Please delete “on the other hand” and begin the sentence with Climate Change. 

43   Also, please reverse “intensity and occurrence”.  The occurrence should appear first because, without the occurrence, you can have no intensity.  This and the following sentences that starts on line 44 can be more clearly written as, “Climate change is expected to alter drought occurrence and intensity.  Increased drought duration in many parts of the world will further challenge drought mitigation strategies [1].” 

46   After [2], I suggest revising to write, “It remains difficult to determine the onset, intensity, and duration of drought in spite of a great number of studies committed to this effort [3].”

49   Replace the word “shortcuts” with “shortcomings”  These words have greatly different meanings.

54-55   Delete “as stated by some authors.”

55   In the sentence that begins with, “One of the problems,” the reader will wonder why there is a problem.  A standardized precipitation index should be based on precipitation data, should it not?  The problem is that this index is being used as an indicator of drought, not as an indicator of relative precipitation.  Therefore, I think you should revise to write, “One of the problems of using the SPI as a drought indicator is that its calculation is based exclusively on precipitation data.  Other variables associated with drought, such as air temperature, are not included.”  

57-58   I suggest you delete this sentence and continue with, “The standardized precipitation evapotranspiration index (SPEI) makes use of both precipitation and air temperature data [2].”

61-62   This sentence is confusing.  Does the SPEI just use precipitation and temperature, or does it also use a climatic water balance of some sort?  If the latter, you should state what that water balance is based on.  Furthermore, you don’t need to state again that it uses temperature.  You’ve already said that. 

66   I suspect there is more than one gap.  If so, please replace gap with gaps.

67   I’m not aware of “groundwater drought.” 

69   Add “the” to write, “One of the main results of this work . . .”

70   Perhaps separate would be a better word than detach.

71-72   I suggest revising to write, “The application of this index could also be applied to assess sustainable management of coastal aquifers.”

73-76   You should cite the source of the recent risk assessment.  Ah, I see that you provide a citation in the following sentence.  I suggest you move that citation to appear in this sentence so it follows the word “needed.”

75   Delete the word “found”

78   I suggest you revise to write, “To avoid any potential risk to the groundwater dependent OUV ecosystems within DNP, it is highly recommended . . .”

83   Replace Taken with Taking

84   I suggest revising to write, “Inform the SPI, SPEI, and GRDI models with historical climatic data from DNP.”

85   Delete “official”

86   Consider replacing “future” with “forecast”

102   You need to show the location of Santa Olalla pond in Figure 1.  Also, I suggest you replace “over” with “only.”

109   When you write, “the pond hydrology,” are you talking about Santa Ollala pond or the larger number of ponds near the dunes?

114   The labels for the colored dots in the map legend are reversed.  “Almonte data” needs to be moved to be next to the black dot and “met. Station” needs to be moved to next to the blue dot.

114   Regarding the location of the met station, it appears to be in the middle of a strange land-use area, based on Google Earth.  What are all those hundreds of rows of what appear to be large, linear plastic bags?  What sort of land use is this?  Are we looking at greenhouses?  This really isn’t important to the manuscript unless there is a certain high-value type of agriculture going on here, particularly if it is water-intensive.

115   A dot is a poor symbol to represent a zone.  I suggest you replace the dot with a shaded area that indicates the zone where future projections apply.

116   The figure displays both yellow shading and a yellow border.  If the groundwater body is represented by the yellow shading, what does the yellow border represent?  Also, what do the shaded areas Marismas de Donana and Marismas indicate?  And what does the green border represent?

143-147   Here is another example of confusing sentence structure and word combinations that are unclear.  I do not know what you mean by “. . . being very common in rainfall studies according to the seasons the use of the SPI for6 months”.  Does this perhaps mean that you integrate data over a 6-month period?  Or do you calculate a 6-month running average?  Basically, how do you aggregate the data over a 6-month or a 12-month period?  And why 6 or 12 months?  You state that you chose a 12-month aggregate because that was better for determining a long-term rainfall pattern.  However, groundwater recharge occurs mostly during your rainy period.  Therefore, I would think a 6-month aggregate would better relate to periods when groundwater recharge occurs.

181   Do you perhaps mean to use the word “protocol” instead of “proposal?”

179   You stated earlier that the GRDI model was especially important to your results.  However, the information provided in this section does not tell the reader how this index works.  How does the soil water balance program work?  How does a runoff threshold relate to this?  Is the threshold of 90 mm related to precipitation?  If so, 90 mm of precipitation summed over how long a period?  An hour?  A day? 

186-187   How can an R package designed to calculate SPEI be used to calculate GRDI?

190-191   You should cite Figure 1 here and indicate the location is shown by the blue dot.

195    You need to either present the Hargreaves equation or at least tell the reader that this is an equation for determining daily values for PET.

196-209   Your results depend greatly on the climate projections from these forecasted results.  You provide the source of these values, but the reader has no indication for their reputation among climatologists.  Anyone can predict future values.  There needs to be some indication of credibility of these predicted values.  Lastly, what do these modeled results indicate?  Will it be warmer.  If so, how much warmer?  Will it be wetter or drier?  By how much?  You need to provide the reader some idea of what type of climate will occur in the future.

236   for Figure 2, I suggest you move panel c so it appears directly below panel a.  This allows the reader to better compare the durations and inflections of dry and wet periods for each of the two models.  Similarly, displaying panel B directly above panel D will allow better inter-model comparison.

252   Ah!  In Figure 3 you display the panels so the reader can better see the differences!  Please rearrange the panels in Figure 2 so they are displayed in the same arrangement as in figure 3.  Also, for panel C, please rescale the plot so the years line up with panels A and B.  As it is, it appears that the period of high recharge in panel C starts before the wet conditions that start around 1995.

261   Figure 4 is not very useful.  The reader, who presumably is interested in groundwater resources, will not care about forecast highs or lows for each year, which is all one can determine in panels B, C, E, and F.  Regarding precipitation, the only thing that is evident is that RCP 8.5 indicates wetter conditions than RCP 4.5.  I suggest you do some trend analysis and summarize all of that seasonal variability into something that is meaningful from a groundwater-recharge perspective.  I suggest you present a plot of trends rather than model output.

274   Now that I see Table 3, I suggest you simply delete Figure 4.  Table 3 presents the same information in a much more reader-friendly manner.

276-284   This paragraph does not say much related to groundwater.  You start by talking about forecast wet and dry periods, which is meaningless unless you compare it to what has been measured.  You do eventually get to that, but then you only partially make the comparison.  You should recast this paragraph from the perspective of the effects of forecast climate change relative to groundwater supply.  A comparison of the number, duration, and severity of dry periods during previous years, relative to the number, duration, and severity of forecast dry periods is what is most important.  Comparing measured versus forecast wet periods is also important related to groundwater recharge. 

281   Why did you only use 29 years of your data that extend from 1995 to 2017?  This same question also applies to the historical data that only extend to 2004 in Table 3.

330   Your statement “Similar results have been found in other regions.” needs supporting citations.

338-343   This is the main result of your paper.  I suggest you highlight this point by placing this important information in a separate paragraph. 

351   This is interesting information, but it is somewhat separate from this current work.  Therefore, you should cite the source of these results before you present them.

398-400   You do not state that the occurrence of drought periods in the future is forecast to be about the same as in the past.  This is an important result that should be included in the Conclusions.  I suggest you place this important sentence just before you state that wet periods are forecast to be reduced.

Author Response

Review of Water 2324522

Overall comments:

QUESTION: This manuscript has the potential to be an important contribution to the literature, but it is poorly presented and difficult to understand.  I was tempted to simply recommend it be rejected, but the main conclusions are important, both for scientists and resource managers.  Therefore, I think it can be substantially modified to make it suitable for publication.

The manuscript gets off to a disappointing start with an abstract that is not useful at all.  It tells the reader very little about the study or even what was found.  All we know is that precipitation was reduced during the second half of the study.  But what did that do to groundwater resources?  The reader learns that recharge rates are reduced, of course.  By how much?  That is what the reader really needs to know.  Based on the title, the paper is about groundwater and drought.  But where is drought mentioned in the abstract?  It isn’t.  You mention three indicators, but you don’t say what the indicators are related to.  You then talk about Outstanding Universal Value ecosystems, assuming that readers know what this is.  I do not.  Later, in the introduction, I find these are ponds and lagoons.  You state in the abstract that these ecosystems are dependent on groundwater recharge.  Knowing these are ponds and lagoons, I think you actually mean that they are dependent on groundwater discharge, not recharge.  For these many reasons, the abstract is very confusing and needs heavy revision.

The rest of the manuscript is similarly confusing due to poor organization and presentation.  I provide numerous suggestions for improvement in the Introduction section, as an indication of what I mean by poor paragraph structure and presentation.  However, I did not do that for the entire manuscript.  The rest of the manuscript also is in need of a thorough edit.

ANSWER: Thank you, we appreciate the comment about the importance for the literature of this contribution. We have tried to implement all the suggestions made by the reviewer. Firstly, as stated by the reviewer, the Abstract have been rewritten in order to appoint all the commentaries. A heavy revision was made. We have explicitly defined all the indexes at the beginning of the Abstract. We have mentioned the total reduction (-13-14%) in precipitation as well as in severely to extreme wet periods (-25-38%) in the new version of Abstract. We also have decided to eliminate from the Abstract the paragraph related to the OUV (Outstanding Universal Value) ponds and lagoons. As the reviewer correctly points out, these ecosystem receive groundwater discharge instead of groundwater recharge, as incorrectly stated in the first version of the Abstract.

Specific comments

Abstract:

QUESTION: The abstract should stand alone.  Please spell out SPI, SPEI and GRDI, or at least explain what these indexes relate to.  I looked up GRDI and I still don’t know what it is.  I doubt it is a Global Retail Development Index, nor is it likely a Gridded Relative Deprivation index.  Just reading the abstract thus far, I have no idea what your manuscript is about.

ANSWER: Thanks. GRDI stands for Groundwater Recharge Drought Index. Among the most important research made in sustainable management of groundwater in the context of Global Change, using meteorological, agricultural, and hydrological indexes, was the development of this index by Goodarzi in 2016. He developed an index to determine groundwater drought. The index was named Groundwater Recharge Drought Index (GRDI). The complete reference: M. Goodarzi, J. Abedi-Koupai, M. Heidarpour and H. R. Safavi, “Development of a New Drought Index for Groundwater and Its Application in Sustainable Groundwater Extraction,” Journal of Water Resources Planning and Management, vol. 142, no. 9, pp. 04016032-1-12, 2016.

QUESTION: You write, “results are consistent with previous studies” but you don’t say what results or what the previous studies pertain to. 

ANSWER: We agree. As the Abstract has been rewritten, in the new version of the Abstract we have eliminated this paragraph.

QUESTION: You state that the most important contribution of your work is the GRDI index, but you never say what that index is, or what it is used for.  Evidently it is related to mitigation measures, but mitigation for what?  This entire abstract is very frustrating to read and should be completely revised.  It tells the reader almost nothing.

ANSWER: We again agree with this comment. As commented above, in the new version of the Abstract we have tried to avoid these several inconsistencies.

Introduction:

QUESTION: 29   I know what a method is, or an approach.  But what is a methodological approach?  I suggest replacing “methodological approach” with “multiple approaches.” 

ANSWER: We agree. We have change “methodological approach” with “multiple approaches.” 

QUESTION: 31   I suggest replacing “acknowledged” with “quantified”.  Also, revise to write “. . . to allow a better understanding of the relative importance of specific hydrological processes.” 

ANSWER: We agree. We have change “acknowledged” with “quantified”.  We have revised and write “ to allow a better understanding of the relative importance of specific hydrological processes.”  In the new version of the MS.

QUESTION: 34-35   “the benefit of using groundwater is evident” is not useful.  Why is it evident?  I suggest replacing with “Densely populated or agricultural coastal areas, where access to fresh surface water commonly is limited, are largely dependent on groundwater.” 

ANSWER: Agree, in the new version of the MS we have used the new phrase: “Densely populated or agricultural coastal areas, where access to fresh surface water commonly is limited, are largely dependent on groundwater”.

QUESTION: 36   You need to provide a citation for your 80 percent value.

ANSWER: In agreement with the reviewer comment, We have provided a citation in the new version of the MS.

QUESTION: 38   What is an outstanding universal value pond or lagoon?  You need to indicate which organization creates or uses this distinction, how it is determined, and cite a source for this term.

ANSWER: Thanks. An OUV pond or lagoon, as defined and categorized by the UNESCO, is an ecosystem that is so exceptional as to transcend national boundaries and to be of common importance for present and future generations of all humanity. As such, the permanent protection of this heritage is of the highest importance to the international community as a whole. A UNESCO Committee defines the criteria for the inscription of properties on the World Heritage List. To be considered of Outstanding Universal Value a site must meet the conditions of integrity (and authenticity) and must have an adequate protection and management system to ensure its safeguarding.

Source: OG Operational Guidelines for the Implementation of the World Heritage Convention (WHC.19/01 - 10 July 2019)

QUESTION: 40   Safe alternative to what?  I suggest “as a safe alternative to surface water”

ANSWER: In agreement with the reviewer comment, we have change the text in the new version of the MS.

QUESTION: 42   Beginning a sentence with “on the other hand” implies that this sentence is presenting an alternate perspective relative to the previous sentence.  It is not.  It is just providing additional information.  Please delete “on the other hand” and begin the sentence with Climate Change. 

ANSWER: Thanks. We have removed “On the other hand” in the new version of our MS as suggested.

QUESTION: 43   Also, please reverse “intensity and occurrence”.  The occurrence should appear first because, without the occurrence, you can have no intensity.  This and the following sentences that starts on line 44 can be more clearly written as, “Climate change is expected to alter drought occurrence and intensity.  Increased drought duration in many parts of the world will further challenge drought mitigation strategies [1].” 

ANSWER: In agreement with the precise reviewer suggestion, we have change the whole paragraph in the new version of the MS.

QUESTION: 46   After [2], I suggest revising to write, “It remains difficult to determine the onset, intensity, and duration of drought in spite of a great number of studies committed to this effort [3].

ANSWER: Agree, in the new version of the MS we have change the phrase in accordance with the reviewer suggestions.

QUESTION: 49   Replace the word “shortcuts” with “shortcomings”  These words have greatly different meanings.

ANSWER: In the new version of the MS we have replaced shortcuts with shortcomings.

QUESTION: 54-55   Delete “as stated by some authors.”

ANSWER: Deleted.

QUESTION: 55   In the sentence that begins with, “One of the problems,” the reader will wonder why there is a problem.  A standardized precipitation index should be based on precipitation data, should it not?  The problem is that this index is being used as an indicator of drought, not as an indicator of relative precipitation.  Therefore, I think you should revise to write, “One of the problems of using the SPI as a drought indicator is that its calculation is based exclusively on precipitation data.  Other variables associated with drought, such as air temperature, are not included.”  

ANSWER: In agreement with the reviewer suggestion, we have changed the whole paragraph in the new version of the MS.

QUESTION: 57-58   I suggest you delete this sentence and continue with, “The standardized precipitation evapotranspiration index (SPEI) makes use of both precipitation and air temperature data [2].”

ANSWER: Deleted the phrase “For that reason, some authors have developed indices related to SPI, but taking also into account evapotranspiration (ET).”

QUESTION: 61-62   This sentence is confusing.  Does the SPEI just use precipitation and temperature, or does it also use a climatic water balance of some sort?  If the latter, you should state what that water balance is based on.  Furthermore, you don’t need to state again that it uses temperature.  You’ve already said that. 

ANSWER: Thanks. The SPEI uses the accumulated difference between precipitation and potential evapotranspiration. SO we have rewritten the phrase to: “The calculation of this index uses the accumulated differences between precipitation and potential evapotranspiration (see Materials and Methods section below). Therefore, it is an excellent method to model the effects of global warming on drought situations.”

QUESTION: 66   I suspect there is more than one gap.  If so, please replace gap with gaps.

ANSWER: In agreement with the reviewer suggestion, we have changed gap for gaps in the new version of the MS.

QUESTION: 67   I’m not aware of “groundwater drought.” 

ANSWER: Groundwater drought is the continuous and extensive occurrence of below average readiness of groundwater. Aquifers are usually replenished with water during the winter months, so groundwater droughts may develop if there is reduced rainfall over one or successive winters. In that sense, is basically the same as “meteorological drought” but focused solely on the groundwater resource. We have explained the concept in the new version of the MS for clarity.

QUESTION: 69   Add “the” to write, “One of the main results of this work . . .”

ANSWER: In agreement with the reviewer suggestion, we have add “the” in the new version of the MS.

QUESTION: 70   Perhaps separate would be a better word than detach.

ANSWER: Correct. In agreement with the reviewer suggestion, we have used “separate” in the new version of the MS.

QUESTION: 71-72   I suggest revising to write, “The application of this index could also be applied to assess sustainable management of coastal aquifers.”

ANSWER: As the referee kindly propose, we have rewritten the phrase in the revised version of the MS.

QUESTION: 73-76   You should cite the source of the recent risk assessment.  Ah, I see that you provide a citation in the following sentence.  I suggest you move that citation to appear in this sentence so it follows the word “needed.”

ANSWER: Good point. We have made that in the new version of the MS.

QUESTION: 75   Delete the word “found”

ANSWER: In agreement with the reviewer suggestion, we have deleted the word “found” in the new version of the MS.

QUESTION: 78   I suggest you revise to write, “To avoid any potential risk to the groundwater dependent OUV ecosystems within DNP, it is highly recommended . . .”

ANSWER: As the referee propose, we have rewritten the phrase in the revised version of the MS.

QUESTION: 83   Replace Taken with Taking

ANSWER: Change was made in the new version of the MS.

QUESTION: 84   I suggest revising to write, “Inform the SPI, SPEI, and GRDI models with historical climatic data from DNP.”

ANSWER: As the referee propose, we have rewritten the phrase in the revised version of the MS.

QUESTION: 85   Delete “official”

ANSWER: Deleted

QUESTION: 86   Consider replacing “future” with “forecast”

ANSWER: Change, replacing “future” with “forecast”, was made in the new version of the MS.

QUESTION: 102   You need to show the location of Santa Olalla pond in Figure 1.  Also, I suggest you replace “over” with “only.”

ANSWER: As the referee propose, we have showed the location of Santa Olalla Pond in the new version of Figure 1. We have also replaced “over” with “only”.

QUESTION: 109   When you write, “the pond hydrology,” are you talking about Santa Ollala pond or the larger number of ponds near the dunes?

ANSWER: Thanks. Change was made in the new version of the MS: “which also influences pond’s hydrology”

QUESTION: 114   The labels for the colored dots in the map legend are reversed.  “Almonte data” needs to be moved to be next to the black dot and “met. Station” needs to be moved to next to the blue dot.

ANSWER: Done

QUESTION: 114   Regarding the location of the met station, it appears to be in the middle of a strange land-use area, based on Google Earth.  What are all those hundreds of rows of what appear to be large, linear plastic bags?  What sort of land use is this?  Are we looking at greenhouses?  This really isn’t important to the manuscript unless there is a certain high-value type of agriculture going on here, particularly if it is water-intensive.

ANSWER: Thanks. The land where the station is placed represent a greenhouse area devoted to berries cultivation (mostly strawberry, blueberry and raspberry).

QUESTION: 115   A dot is a poor symbol to represent a zone.  I suggest you replace the dot with a shaded area that indicates the zone where future projections apply.

ANSWER: We appreciate the reviewer suggestion, but the dot represent the point where the downscaling is taken from. So we still think that a point represents better the coordinates than a shaded area.

QUESTION: 116   The figure displays both yellow shading and a yellow border.  If the groundwater body is represented by the yellow shading, what does the yellow border represent?  Also, what do the shaded areas Marismas de Donana and Marismas indicate?  And what does the green border represent?

ANSWER: The yellow border represent the Almonte-Marismas detrital aquifer. This aquifer was divided into 5 groundwater bodies: Marismas de Doñana (green shaded), Marismas (pink shaded), Manto Eólico Litoral de Doñana (yellow shaded), Rocina and Almonte. The green border represents the limits of Doñana National Park. We have added this intofrmation (not mentioned) in the new version of the MS.

QUESTION: 143-147   Here is another example of confusing sentence structure and word combinations that are unclear.  I do not know what you mean by “. . . being very common in rainfall studies according to the seasons the use of the SPI for6 months”.  Does this perhaps mean that you integrate data over a 6-month period?  Or do you calculate a 6-month running average?  Basically, how do you aggregate the data over a 6-month or a 12-month period?  And why 6 or 12 months?  You state that you chose a 12-month aggregate because that was better for determining a long-term rainfall pattern.  However, groundwater recharge occurs mostly during your rainy period.  Therefore, I would think a 6-month aggregate would better relate to periods when groundwater recharge occurs.

ANSWER: Thanks. We have rewritten the phrase in order to improve clarity in the new version of the ms:

QUESTION: 181   Do you perhaps mean to use the word “protocol” instead of “proposal?”

ANSWER: Acknowledgements. Change was made in the new version of the MS: “protocol” was used instead of “proposal”.

QUESTION: 179   You stated earlier that the GRDI model was especially important to your results.  However, the information provided in this section does not tell the reader how this index works.  How does the soil water balance program work?  How does a runoff threshold relate to this?  Is the threshold of 90 mm related to precipitation?  If so, 90 mm of precipitation summed over how long a period?  An hour?  A day? 

ANSWER: We appreciate the referee question about the methodology. The section GRDI was improved in order to a better understanding of the process involved in the calculation of this novel index.

QUESTION: 186-187   How can an R package designed to calculate SPEI be used to calculate GRDI?

ANSWER: We appreciate the referee question about the R package.. In the new version of the MS, the paragraph goes “To use this package, instead of inserting precipitation as the input time series, we use the recharge values obtained by the implementation of the TRASERO code at a daily scale and afterwards upscaled to obtain month recharge values .”

QUESTION: 190-191   You should cite Figure 1 here and indicate the location is shown by the blue dot.

ANSWER: Figure 1 is cited in the paragraph: “We have used 16 different models for RCP 4.5 and RCP 8.5 scenarios from the IPCC for the closest area to our study site (see Fig. 1).”

QUESTION: 195    You need to either present the Hargreaves equation or at least tell the reader that this is an equation for determining daily values for PET.

ANSWER: We appreciate the referee comments and we have modified the phrase in the new version of the MS: “Daily data were aggregated to monthly values. With the package SPEI in R Studio [15], potential evapotranspiration (PET) data was calculated using Hargreaves equation. The Hargreaves equation is a temperature based method and it is used as a representative expression for daily values of PET:

PET (mm/day) = 0,0135 (tave + 17,78) Rs

Were:

tave = average temperature, ºC

Rs = incident solar radiation, mm/day

Rs = R0KT (tmax – tmin)0.5

R0 = extraterrestrial solar radiation, MJules/m2/day

KT = empirical coefficient (0.162 for interior and 0.19 for coastal areas)

tmax = maximum temperature, ºC

tmin = minimum temperature, ºC”

QUESTION: 196-209   Your results depend greatly on the climate projections from these forecasted results.  You provide the source of these values, but the reader has no indication for their reputation among climatologists.  Anyone can predict future values.  There needs to be some indication of credibility of these predicted values.  Lastly, what do these modeled results indicate?  Will it be warmer.  If so, how much warmer?  Will it be wetter or drier?  By how much?  You need to provide the reader some idea of what type of climate will occur in the future.

ANSWER: Thanks. In the new version of the MS we have indicate the credibility of the predicted values. As indicated, the EUROCORDEX project is a reliable scientific project. In the new version of the MS, we have included more information about this matter, as indicated by the reviewer.

QUESTION: 236   for Figure 2, I suggest you move panel c so it appears directly below panel a.  This allows the reader to better compare the durations and inflections of dry and wet periods for each of the two models.  Similarly, displaying panel B directly above panel D will allow better inter-model comparison.

ANSWER: Thanks. We have decided to maintain the order of the figures. The duration and inflection of SPI and SPEI dry and wet periods are very similar. Besides, in figure 3 we decided to display the panels that way, as the referee points out.

QUESTION: 252   Ah!  In Figure 3 you display the panels so the reader can better see the differences!  Please rearrange the panels in Figure 2 so they are displayed in the same arrangement as in figure 3.  Also, for panel C, please rescale the plot so the years line up with panels A and B.  As it is, it appears that the period of high recharge in panel C starts before the wet conditions that start around 1995.

ANSWER: Thanks, we have tried to amend the figure accordingly.

QUESTION: 261   Figure 4 is not very useful.  The reader, who presumably is interested in groundwater resources, will not care about forecast highs or lows for each year, which is all one can determine in panels B, C, E, and F.  Regarding precipitation, the only thing that is evident is that RCP 8.5 indicates wetter conditions than RCP 4.5.  I suggest you do some trend analysis and summarize all of that seasonal variability into something that is meaningful from a groundwater-recharge perspective.  I suggest you present a plot of trends rather than model output.

ANSWER: See question and answers to Q274 below.

QUESTION: 274   Now that I see Table 3, I suggest you simply delete Figure 4.  Table 3 presents the same information in a much more reader-friendly manner.

ANSWER: As the reviewer suggest, we have deleted figure 4 from the MS in our revised version.

QUESTION: 276-284   This paragraph does not say much related to groundwater.  You start by talking about forecast wet and dry periods, which is meaningless unless you compare it to what has been measured.  You do eventually get to that, but then you only partially make the comparison.  You should recast this paragraph from the perspective of the effects of forecast climate change relative to groundwater supply.  A comparison of the number, duration, and severity of dry periods during previous years, relative to the number, duration, and severity of forecast dry periods is what is most important.  Comparing measured versus forecast wet periods is also important related to groundwater recharge. 

ANSWER: Correct, we agree with your question, and have made changes in the paragraph accordingly. We have compared the number, duration, and severity of dry periods during previous years, and we have related them to the number, duration, and severity of forecast dry periods. Moreover, we have compared measured vs. forecast wet period.

QUESTION: 281   Why did you only use 29 years of your data that extend from 1995 to 2017?  This same question also applies to the historical data that only extend to 2004 in Table 3.

ANSWER: In the first case we decided to use a time spam of 29 years to compare with a similar range from  2030 to 2060. The historical data only extend to 2004 as we decided to use a time series less affected by the increment in temperatures due to climate change.

QUESTION: 330   Your statement “Similar results have been found in other regions.” needs supporting citations.

ANSWER: Thanks. New text and references were added in the new version of the MS.

QUESTION: 338-343   This is the main result of your paper.  I suggest you highlight this point by placing this important information in a separate paragraph. 

ANSWER: We agree with the comments of the referee. In the new version of the MS we have placed this whole part in a separate paragraph: “Although there was evidence from modelling that climate change could alter the hy-drology and ecology of the Doñana in the future, the result of the present study does not pose evidence of an increment of droughts. Our results show a similar frequency of droughts, for the same period length of 30 years two long two long periods of drought of four and five years. The significant difference is obtained for the wet periods. The periods with SPI severely to extremely wet were 25% lower in the future predictions with respect to the historical time series. In addition, the periods with SPEI severely to extremely wet were 38% lower in the future predictions with respect to the historical time series. This is precisely the reality of nowadays, as since year 2010 there has not been any wet year. This phenomenon is resulting water resources scarcity in Doñana area, as well as it happens in the drought periods. The significant increment in mean and maximum temperatures, that would result in a subsequent increment of ET in the area, reduces recharge rates and adds a negative effect to the OUVs of the park.”

QUESTION: 351   This is interesting information, but it is somewhat separate from this current work.  Therefore, you should cite the source of these results before you present them.

ANSWER: Thank you. In fact, this is a good example of “grey” literature, i.e., academic reports or technical information that it is normally written in Spanish (in the case of DNP) and that it is difficult or nearly impossible to read elsewhere. Sometimes, the information provided in such reports are of enormous importance to understand the hydrological processes taking place at one particular pond or sometimes in the whole aquifer. In this particular case, this Bachelor Thesis was conducted by one of the coauthors of this work, Dr. Miguel Rodríguez-Rodríguez, who will be willing to attach the Thesis. This will be made - as supplementary material - in the new version of the MS.

QUESTION: 398-400   You do not state that the occurrence of drought periods in the future is forecast to be about the same as in the past.  This is an important result that should be included in the Conclusions.  I suggest you place this important sentence just before you state that wet periods are forecast to be reduced.

ANSWER: Thanks. In fact, the referee is right. Once stated that the analysis of drought and pluvial events indicates two important drought periods and two significant pluvial events, in the new version of the MS we have included the new phrase: “The occurrence of drought periods in the future is forecast to be about the same as in the past”

We acknowledge the insight of the reviewer and are convinced that their suggestions have improved substantially the new version of our work.

Please find attached the revised version (NEW VERSION MS water-2324522.docx)

Reviewer 2 Report

The paper submitted by  Montes Vega et al. is a study on the climatic varibility of the Doñana coastal area, but no information is provided on groundwater or aquifer despite the whole paper is focusing on the climate change effects on groundwater resources. For instance, the introduction is mentioning many times groundwater resources, but the materials and methods only used metereological data to calculate standard indices for the hydrological budget. No piezometric maps are given no monitoring of groundwater levels during time is provided no clues about the hydrogeological units that for the unconfined and confined aquifers. In fact, the conclusions are not aligned with the title and just provide information on atmospheric temperatures that are forecasted to increase and consequently the potential evapotranspiration rates.

Considering the flaws highlighted above I cannot recommed this paper for publication in Water.

Author Response

* See modifications in the reviewer 1 reply